



# Boundary|Time|Surface: Art and geology meet in Gros Morne National Park, Newfoundland, Canada

Sydney A. Lancaster[1], John W.F. Waldron[2]

[1]Edmonton, Alberta, T6E1G6, Canada
[2]Earth and Atmospheric Sciences, University of Alberta, Edmonton, T6G2E3, Canada

Sydney A. Lancaster: ORCID: 0000-0002-5843-3947
John W.F. Waldron: ORCID: 0000-0002-1401-8848

*Correspondence to*: Sydney A. Lancaster (sydneylancaster@ualberta.net)



**Abstract.** *Boundary|Time|Surface* was an ephemeral sculptural work created to interrogate the human practice of dividing the Earth for social, political, scientific and aesthetic reasons. The 150-metre-long work comprised a fence of 52 vertical driftwood

poles, 2-3 metres tall, positioned along an international boundary stratotype at Green Point, Newfoundland, Canada, separating Ordovician from Cambrian strata.

 Geology has as its basis the establishment of limits and boundaries within the Earth. Pioneers of geology defined the periods of the geologic timescale with the intent of representing natural chapters in Earth history; from their colonialist perspective, it was anticipated that these would have global application. Since the mid-20th century, stratigraphers have attempted to resolve

the resulting gaps and overlaps by establishing international stratotypes.

Artists creating work in dialogue with the land and environment have taken a wide range of approaches, from major, permanent interventions to extremely ephemeral activities, some of which echo practices in geological fieldwork. Because it was constructed in a national park, *Boundary|Time|Surface* was designed to have minimal impact on the environment; the installation was constructed by hand from materials found on site in one day, on the falling tide. During the remainder of that

tidal cycle, and those following, the fence was dismantled by wave and tidal action. This cycle of construction and destruction was documented in video and with time-lapse still photography.

Exhibitions derived from the documentation of ephemeral works function as translations of the original experience, offering an extended opportunity for members of the public to experience aspects of the original work and its context. Two exhibitions of artwork derived from *Boundary|Time|Surface* have provided opportunities for several thousand members of the public to

interact with a range of visual media directly, both as aesthetic objects, and as sources of information regarding the geological and socio-political history of the site. A limited-edition book published in September 2019, to accompany an exhibition of the work, has extended the reach of the project further.





## 1 Introduction

An outdoor ephemeral art installation, *Boundary|Time|Surface* provided a brief, site-specific opportunity for viewers to contemplate the human experience relative to the enormity of geological time, and the fragile and arbitrary nature of human-defined boundaries. The installation was created on the west coast of the large island known as Ktaqumkuk by the Indigenous Mi'kmaq, as Terre Neuve by its early French settlers, and as Newfoundland by its anglophone population within the political structure of modern Canada. In this paper, we describe the social, artistic, and scientific context within which the work was

constructed. We then outline the construction and destruction of the work itself, and the photographic and videographic methods that captured these processes. Ephemeral artworks have, by their nature, a limited audience; to communicate their 'findings' for projects such as these, artists must rely upon various methods of documentation, both as lasting records of the works' existence, and as tools with which to extend the reach of the original project. We here describe the ways in which the reach of the original ephemeral work has been, and continues to be, extended so as to inform a larger audience.

## 2 Background


### 2.1 Social and cultural context

There is a common human desire for a sense of permanence in the world; humans retain a psychological and emotional attachment to the notion that both ideas and objects such as walls, borders, and boundaries are in some way *permanent* (Nail, 2016, p.6–7) despite a range of indications in the everyday world to the contrary. Boundaries are perceived as enduring marks

made by humanity, that simultaneously prove our importance to the history of the planet, and assure us that there are fixed points upon which we can feel assured of ongoing security, outside the passage of time and (sometimes dramatic) socio-political change (Wood, 2019, p.80–81).

Individuals speak colloquially of "crossing the line" or "drawing a line in the sand" or even "invading personal space" to mark various limits and points of transgression. So too, social forces and political entities create borders, erect boundary markers,

declare and define limits: limits for time, limits for physical space and movement, and limits that serve to identify places, things, and people, and that privilege certain ways of knowing over others. While they are often viewed as fixed expressions of verifiable 'truth,' the information within these containers, like the containers themselves, is arbitrary. Both container and information therein are subjective creations expressing power relations in physical and temporal space *(*Zeller, 2000). *Boundary|Time|Surface* was created as an intervention within a specific landscape to address these ideas.

### 2.2 Artwork in the landscape


"Earth art", "land art", and "environmental art" are terms that cover a range of site-specific artistic works that arose as part of a shift toward Conceptual Art in the 1960s, in part as a response to the commercialization of traditional forms of art displayed in museums and galleries. In the United States, this style of art-making was pioneered by Robert Smithson, whose *Spiral Jetty* (1970) incorporated over 6000 t of basalt and earth moved from industrial wasteland on the shore of Great Salt Lake, to form

a spiral in the lake's water that measured 4.4 m by 460 m (The Art Story, 2015). As a work that responded directly to the geology and specific details of the landscape (Smithson, 1996, p.143–152), *Spiral Jetty* also had an unexpected relationship to the growing awareness of anthropogenic environmental change; the installation was submerged due to rising lake levels in 1972, and its subsequently re-emerged, covered with a layer of evaporite deposits, during lake-level fall in 2002 (Casey, 2005; Hopkins, 2000).

More explicitly connected with the phenomena studied by Earth science is James Turrell's *Roden Crater* (begun 1972), a massive structure of connecting rooms and tunnels built as a naked eye observatory into an extinct volcanic cinder cone in the





Painted Desert in Northern Arizona USA(Cook, 2010). This artwork, still under construction, has thusfar included the movement of ~$10^6$ m$^3$ of earth. When completed, the project is planned to contain 21 viewing spaces and six tunnels, a temple-like space allowing visitors to experience celestial events occurring at various times (Fredricksen, 2002; Turrell, 2020).

In contrast to the American tradition of land art, marked by major interventions in the landscape, an alternative land art tradition arose in the UK. One of the earliest practitioners in this tradition, Richard Long, has been characterized as a "walking artist." Beginning in the 1960's, Long created a series of artworks based in multi-day walks through the landscape (Dapena-Tretter, 2014), and the documentation of these journeys in photographs, maps, and text works  - a practice with some parallels in geological mapping, practitioners of which have long been expected to walk long distances over rugged ground during the

collection of geological data. Long also works in site-specific sculptural installation, often using natural materials found on location in the landscape (Long, n.d.).

Likewise, Scottish artist Andy Goldsworthy has created site-responsive works of various sizes throughout his career. Crucially, many of his artworks are made from ephemeral, organic materials (Tufnell, 2006); the natural life cycle of the materials at hand is intrinsic to the work itself, as is its eventual disappearance (Gooding, 2002, p.21–23; The Art Story, 2018). Several of

Goldsworthy's projects have involved explicit reference to Earth science phenomena (Lubow, 2005).  One work, created from driftwood on the shores of the Bay of Fundy at Fox River, Nova Scotia, Canada, was designed to evoke the movement of a whirlpool – a phenomenon which occurs in the Bay itself, and that Goldsworthy observed in a pool of water close to the shore. The work itself was lifted up essentially intact inside the pool by the incoming tide, and spun slowly around as it was pushed upstream (Goldsworthy, 2000, p.114–117). Similarly, Goldsworthy has created a number of works that echo the shape of

meandering rivers; these have been carved out of sand in various beaches, drawn through snow strewn on ice covered rivers, sculpted out of packed sand or clay, and drawn on a range of surfaces with water (Goldsworthy, 2000, p.74–77, 84–95, 122–129).

Artistic creations such as these help to bring awareness of the phenomena studied by Earth scientists to an audience that would not otherwise undertake formal training in geology or related subjects, and introduce ways of thinking about the Earth that are

different from those employed by scientists and science students. Thus, work of this type offers opportunities to bridge the worlds of scientific research, artistic practice, and the general public, by offering visual imagery and material objects that refer to natural processes and scientific concepts in ways that can provoke metaphorical connections and a deeper understanding of perspectives on the natural world.

### 2.3 History of the Cambrian-Ordovician boundary

*2.3.1 Historical Pioneers*

The recognition that strata record a succession of events in geological history was a product of Renaissance natural philosophy (e.g. Steno, 1669). An important proponent of the relationship between strata and time was Scottish agricultural scientist, geologist, chemist, physician and natural philosopher James Hutton, whose "Theory of the Earth" (1788, p.304) ended with the famous quotation:

*But if the succession of worlds is established in the system of nature, it is in vain to look for any thing higher in the origin of the earth. The result, therefore, of our present enquiry is, that we find no vestige of a beginning -- no prospect of an end.*

An implication of Hutton's work was that the landscape in which he lived could be divided according to the succession of the underlying rock units in Earth history. It must be noted too, that Hutton saw this geological evidence of ongoing process as part of a larger system "…particularly adapted to the purpose of man, who inhabits all its climates, who measures its extent,

and determines its productions at his pleasure" (1788, p.294–295). However, this subdivision was not realized until the early




19th century, in the work of English canal engineer William Smith, who produced what was arguably the first geological map (Smith, 1815; Winchester, 2002). Smith came from a middle-class background as a canal engineer; subsequent history of his exploitation and bankruptcy at the hands of a moneyed establishment, and his eventual recognition and rehabiliation, is well described by Winchester (2002). Smith marked the outcrop extent of strata in different colours, separated by boundaries

drawn on a map of the landscape. The three-dimensional character of the underlying units was represented in the construction by Smith of cross-sections, and even in the colouring of the map, highlighting with shading the steep slopes created by certain erosion-resistant units.

Parts of Britain remained undivided on Smith's map, particularly the areas underlain by older strata, now understood to be more highly deformed, in Wales and Scotland. The challenge of extending Smith's paradigm to these areas was taken up by a

number of 19th century geologists, notably Adam Sedgwick and Roderick Murchison. Adam Sedgwick was the younger son of a clergyman who was elected to the Woodwardian chair of geology at Cambridge University in 1818 largely as a result of his friends' concern to provide him with a source of income (Clark and Hughes, 1890). He became a celebrated lecturer whose students included, in 1831, the young Charles Darwin. Murchison came from a more privileged background and took up geology as a pastime following his demobilization from the British army at the end of the Napoleonic wars (Geikie, 1875).

The two met at the Geological Society of London and worked together in extending the mapping of British strata into older units not effectively separated on Smith's (1815) map. Their work on the geology of Wales and the Welsh borders (Sedgwick and Murchison, 1836) established the Silurian and Cambrian systems, respectively in the Welsh Borders and in central parts of the Welsh basin.  However, the two quarrelled over the boundary between the two systems, leading to their estrangement during the last years of Sedgwick's life. The conflict was not resolved until after Sedgwick's death, when Charles Lapworth

(1879) proposed the establishment of the Ordovician system, broadly encompassing the strata overlapped by Sedgwick and Murchison, between an unconformity at the base of the Arenig Series and another unconformity at the base of the Llandovery Series.

From a modern geological point of view, the controversy between Sedgwick and Murchison appears futile. However, at the time, the periods of the geological time scale were regarded as natural chapters in a cohesive Earth history, separated by major

upheavals and even global catastrophes. Perhaps informed by colonialist perspectives prevalent at the time (Chandna, 2009; Harrison, 2005; Zeller, 2000), early geologists expected boundaries defined in Europe to be traceable all over the world. As a result, many of the boundaries introduced in the early 19th century were placed either at unconformities (with the unsatisfactory result that a span of geologic time was unrepresented at the boundary) or at major facies changes (with the result that faunal changes represented environmental, local events rather than global, evolutionary, changes). For example, the boundary

between Silurian and Devonian systems, placed in Shropshire at the Ludlow Bone Bed, marks the highest occurrence of graptolites in England and Wales, where conditions changed from marine to largely non-marine (Sedgwick and Murchison, 1839). The disappearance of graptolites came to be widely used as a boundary between the two systems, but by 1960 it was clear that graptolites persisted in central Europe and North America well after their disappearance in Britain (Becker et al., 2012), and rocks were being characterized as Silurian in these locations that were clearly younger than Devonian rocks in the

area of the original definition.

### 2.3.2 The stratotype concept in the 20th Century

The gaps and overlaps in the geological time scale continued to cause controversy into the 20th century, when the development of stratigraphic codes (e.g. Hedberg, 1976) led to the introduction of the idea of *type sections* or *stratotypes*: designated localities where stratigraphic units were formally defined, and with which other stratigraphic sections would be correlated.

The benefit of this approach is that it separates the business of *definition* of a unit, which is (ideally) done once, from the business of *correlation* which is subject to uncertainty, because of the incompleteness of both the geological record and the





data collected by geologists. The selection of stratotypes is arbitrary in principle, but in practice has to be conditioned by the correlation criteria that are to be used. Thus, stratotypes for units that represent geologic time need to be placed in successions that contain markers that have a wide distribution and which record changes that are as synchronous as possible (typically the

first appearance datum of a new species of marine planktonic or nektonic fossils). Typically, they are placed in sections that have been intensively studied (Fig. 1).

The first of the boundaries to be redefined was the Silurian–Devonian boundary, which was redefined at Klonk in what is now the Czech Republic (Martinsson, 1977), at a point in a continuous succession of graptolitic shales, somewhat younger than Sedgwick and Murchison's (1839) position. Debate over the choice of other stratotypes marking the boundaries between

Phanerozoic systems has continued through the succeeding decades, and most of them have now been defined by the International Commission on Stratigraphy (ICS) (e.g. Gradstein et al., 2012).

### 2.3.3. Green Point and the establishment of the Cambrian-Ordovician boundary

Lapworth's Cambrian–Ordovician boundary was placed at an unconformity within the successions of North Wales, at the base of the Arenig (now Floian) Series, but 20th century opinion, summarized in Bassett and Dean (1982), favoured a somewhat

lower position, at or close to the first appearance of planktonic graptolites, near the base of the Tremadocian Series. During the following years, conodonts were found to be more cosmopolitan in their distribution than graptolites, and came to be favoured for use in the definition of the boundary. The west coast of Newfoundland/Terre Neuve/Ktaqamkuk, in Canada, exposes the Cow Head Group, a succession of Cambrian to Ordovician slope sedimentary rocks formed on the margin of the Paleozoic Iapetus Ocean. The succession was initially mapped at Cow Head by Whittington and Kindle (1963), who showed

that it contained the Cambrian-Ordovician boundary (as it was then understood). Correlation between the multiple sections along the coast was achieved by James and Stevens (1986) who identified a section at Green Point (Fig. 2), a location in Gros Morne National Park, as the most distal part of the slope succession. The succession contains fossils from four different biostratigraphically useful fossil groups: conodonts, trilobites, graptolites, and radiolarians. The international global Global Boundary Stratotype Section and Point (GSSP) for the Cambrian-Ordovician boundary was defined at Green Point in 2001 at

a point in the section described by Cooper et al. (2001), the middle of their bed 23, at the base of the *Iapetognathus fluctivagus* conodont Biozone. This was the location chosen by us for *Boundary|Time|Surface* in 2014.

### 3 Implementation: Boundary|Time|Surface

#### 3.1 Preparation

*Boundary|Time|Surface* was developed and executed during a 5-week Artist's Residency (Art in the Park) at Gros Morne

National Park in 2014. As this artwork was being created in a Canadian National Park, it was particularly important to minimize any potential environmental impact the work's creation might have. We chose to use only natural materials found on or close to the site. This was an active decision appropriate to both the Parks Canada mandate and regulations, and appropriate to the underlying approach to the work; the goal was to leave little to no trace of our intervention in the landscape over the long term.

The initial task in preparing for the installation was establishing the location of the Cambrian-Ordovician boundary at Green

Point, per the description in Cooper, Nowlan, and Williams' (2001) paper. Having established the location of Bed 23 – designated as the Cambrian–Ordovician boundary – we traced the bed out onto the wave-cut platform to the low-water mark, in order to establish the extent of the work. Once the physical site and dimensions had been established, the authors spent 3 weeks gathering materials for the creation of the work.  Fifty-two driftwood logs and poles were collected at Green Cove, ~320 m from Green Point, and carried to a designated collection site at Green Point. These included both naturally weathered

small tree trunks, and poles that bore evidence of former use in wharves, fish flakes (structures for drying fish), and other



artifacts of the fishing industry, in the form of nails and dressed surfaces. Approximately 450 cobbles, weighing between 2 and 10 kg, were gathered by hand from the shoreline at Green Point, and dispersed in cairns at roughly equal intervals along the along Bed 23. These cairns of stones would be the basic support for the upright driftwood poles, to form a 'fence' along the C-O boundary.

**3.2 Installation day**

The work was created over a single four-hour period during the falling tide on June 22, 2014, beginning at 09:30 am. 8 people collaborated in the construction of the work: Lancaster, Waldron, and 6 additional volunteers. When complete, the work was ~150 m in length, and the poles, spaced ~3 m apart, ranged in height from ~1.8 to ~2.4 m. Low tide occurred at 12:57. The work was completed at approximately this time, with the installation of the most seaward of the 52 poles, and was observed by the installation team and visitors to the site over the course of the day (Fig. 2). Evidence of the work's existence remained at the site for approximately 48 hours: 34 poles had been felled by the incoming tide by sunset on June 22nd, 2014; 5 remained standing on the morning of June 23rd; and one remained on June 24th.

**3.3 Documentation of the installation**

The installation was documented in a time-lapse photographic sequence, video, and individual still photographs over the course of the entire construction day, from before the beginning of construction until the last daylight at ~21:00. The time-lapse sequence was recorded from the shoreline, near to the location of the first pole; it comprises 4023 individual images taken at 10 s intervals, and represents the most detailed document of the lifespan of the installation (Fig. 4). Video was captured for two cameras; one positioned at the clifftop (Fig. 2), and one hand-held, which was placed in various locations on the shore throughout the day. In addition, Lancaster used a head-mounted video camera to capture a personal view of the installation as it was constructed over the course of the 4-hour installation period (Fig. 5). Video captured by Lancaster also recorded discussions between Lancaster, Waldron, and the volunteers regarding the process of construction, as it pertained to the geology of the area and various choices and complications that arose over the course of building the work. Over 400 still photographs of the work from various vantage points were also captured over the course of the day, and also on the following morning from the clifftop, to record the remains of the installation after the high tide cycle of the previous night. The cliff face, wave-cut platform, and surrounding landscape were also extensively documented in video and still photographs in the days both before and after the installation was created; this documentation included approximately 2 hours of raw video and an additional 550 still images.

**3.4 Related site-specific work**

During the period after the construction and dissolution of the main installation, there was an opportunity to create smaller site-specific works at and around the Cambrian–Ordovician boundary stratotype. The principal materials for these came from a rock material that contrasted with the limestone and fissile shale that forms the bedrock at Green Point. This material was *pencil slate* from a location further inland within Gros Morne National Park, where deformation of Ordovician shale has imparted a fabric –slaty cleavage – causing the rock to split most easily along planes at a high angle to the original bedding, while still retaining some of its bedding-parallel fissility. As a result, the rock splits into pencil-like rods, which were used to build smaller scale sculptures along the Cambrian–Ordovician boundary. These were documented photographically and formed an addition to the published work. Examples are shown in Fig. 6.



## 4 Subsequent work

When creating site-specific ephemeral artworks, finding ways to increase the audience for these works is an immediate and ongoing challenge. There is no way to replicate the original work; part of its impact is the direct relationship of the artwork to
its location in the environment. Moreover, the original installation may no longer exist in recognizable form, thus eliminating a tangible reference point for a viewer to seek out a personal experience of the original artwork. The documentation of the work complicates reception further, as it is by its nature an 'edited version' of what once existed: these records are captured from particular vantage points, and thus can never convey the entire experience of the original installation, nor its context. Consequently, using the documentation of site-specific work for increasing the audience for an ephemeral work amounts to an
act of translation. Despite these limitations, however, the presentation of this project in a range of contexts has offered a variety of opportunities to stimulate reflection and the transmission of ideas and information that are not immediately available to the viewer at the site of the original. In particular, the collected visual materials allow the simultaneous presentation of different types of information and scales of time, providing opportunities for the viewer to create connections between ideas and images, and to contemplate those connections at their own pace (O'Rourke 2016: 38-39). For this project, the authors have employed
3 methods for expanding the reach of *Boundary|Time|Surface*: gallery exhibitions; talks; and a book.

### 4.1 Gallery exhibitions

Two exhibitions of work arising from the original installation project have been completed at the time or writing, one in Newfoundland and one in Alberta, Canada. The Newfoundland exhibition took place in the Gros Morne National Park Discovery Centre, a facility incorporating an art gallery as well as a series of exhibits about the natural environment and history
of the Park itself. For this exhibition, we designed a brief introductory panel and two didactic panels in English and French (Fig. 7), appropriate to the museum setting, to outline the history and scientific significance of the Green Point section. The second exhibition took place in the Art Gallery of St. Albert, AB. For the art gallery setting only the brief introductory panel was used.

For gallery presentations of *Boundary|Time|Surface*, it was vital that work derived from the documentation of the original
installation conveyed a sense of different scales of time evident in the site in as many ways as possible; video was an ideal tool for addressing this concern. Multi-panel video installations were developed, that incorporated several clips, some shot in 'real time', some in time-lapse, and some in slow-motion. In these installations, time operates at different scales on different screens, emphasizing the experience of scales of time simultaneously present at the original site: clock time, the diurnal cycle, the tide cycle, human historical time, and geological time. The presentation of the main video works as projection-mapped multi-panel
installations also emphasized shifts in physical scale, and referred to the spatial, sculptural nature of both the landscape and the installation itself. Video clips ranged from long-distance shots, incorporating large sections of the beach and cliff, to close-up segments as the incoming tide covered the lens of the camera, revealing the range of aquatic life below the surface. In each exhibition, the video installation has been re-mapped to the specific gallery environment, further reinforcing the specificity of experience in both the original site and in the gallery (Fig. 8 a).

Photographs and gel-transfer prints of photos, maps, and text were used to suggest the range of information that has been gathered about Green Point over time. These different ways of understanding the place - a seismic reflection profile shot in the adjacent Gulf of St Lawrence, stratigraphic columns of the cliff surface, google maps, topographic maps, images of conodont fossils, photos of the landscape – are discrete methods of interrogating the significance of the site, but each taken in isolation provides only an imperfect understanding. As in the video, these printed images explored and disrupted both physical
and temporal scales; images of the landscape and cliff face were presented on a range of semi-transparent and transparent media in both panoramic and close-cropped formats, and the scales of the images were not correlated to each other, or to a





base map (Fig. 8 c-d). For example, a work titled *167 Lifetimes* (Fig. 8 e) presented an enlarged image of shale and limestone beds on the shore, which were 10-12 cm across in outcrop; the printed image is ~76 cm square, and has a series of 167 tick marks drawn over it in glass paint. Each tick mark represents one 80-year human lifespan; the total duration – about 13000

years – is our rough estimate of the length of time it would have taken for these beds to be deposited, based on the stratigraphic work of James and Stevens (1986) and the time scale of Cooper et al. (2012).

A multi-panel installation printed on translucent silk panels allowed viewer interaction; this work presented a photograph of the original installation of driftwood poles, divided into sections, and presented in 3D space, allowing enough room for visitors

to connect to the experience of walking along and between the line of poles at Green Point, and second, to emphasize the ephemeral nature of the original installation, and by extension, that of all human-made borders and boundaries. The lightweight silk organza offers transparency and movement, suggesting a mirage or dream that the viewer can pass through (Fig. 8 b).

### 4.2 Public presentations

Another means of extending the reach of the project has been through slide and video enhanced talks to a wide range of

audiences. In addition to the poster presentation at the European Geosciences Union in 2015, in the last six years, Lancaster and Waldron have given 11 presentations in total on *Boundary|Time|Surface*: four to general audiences in Newfoundland, Nova Scotia, and Alberta, and seven to scientific/academic audiences in Alberta, Nova Scotia, Newfoundland, and Québec. Audiences ranged in size between roughly 20 and 35 for each of the general-audience presentations, and between 25 and 50 for the scientific/academic audiences (see Table 1).

### 4.4 Book: Boundary|Time|Surface - a record of change

In addition to talks and exhibitions, a limited-edition book on the project was published in 2019 to coincide with the second gallery exhibition of work derived from the original project. The print run was limited to 200 copies, signed and numbered, printed in full colour. *Boundary|Time|Surface - a record of change* (Lancaster and Waldron, 2019) contains essays on the project from art historian and curator Melinda Pinfold (2019), an essay from Waldron (2019) on the history of geology, and

an essay and poetry from Lancaster (2019b, 2019a) reflecting on her development and execution of the project. In addition, the book presents a wide range of visual material, including photographs of Green Point, the original *Boundary|Time|Surface* installation, and work presented in galleries. The book is held in private collections in Alberta, Nova Scotia, Newfoundland, Québec, and two copies are stored with the National Library and Archives of Canada. Remaining copies of the book are available for international purchase via Lancaster's website, through the Art Gallery of Alberta gift shop, and the Atlantic

Geoscience Society.

### 4.5. Exhibition attendance and feedback

Overall attendance at the Discovery Centre from May 20, 2016 to October 10, 2016 was 34,787 people; while no separate attendance records were kept specifically for the art gallery at the Parks Canada Discovery Centre, Parks Canada assumes that the overall visitor numbers for a season reflect exhibition visits as well (R. Hingston, Parks Canada, personal communication

2019). A total of 390 people signed the guest book left in the gallery (Appendix I). The highest proportion of visitors were Canadian, and included individuals from all provinces and two of the three territories. There were a number of visitors from several states in the US, and several from Western European countries, including France, Switzerland, Austria, and Spain, and England. There were also visitors from Australia, NZ, British Virgin Islands, Thailand, and China. Comments were positive, but tended to be of a general nature, in part due to the limited space afforded for recording responses to the exhibition.





Nonetheless, there were some comments that indicated that people spent time with the exhibition, and were responding to the more abstract ideas presented therein (Table 2).

For the second exhibition of work, at the Art Gallery of St. Albert in the City of St. Albert, AB, approximately 1000 people visited the exhibition between September 5 and November 2 2019; an additional 150 attended the Opening Reception. Response to the exhibition was positive, and the curator noted that:

*"Any gallery patrons who had previously visited Gros Morne National Park instantly recognized it as the site of your works. Many visitors enjoyed the blending of art and science in your exhibition, and spent a long time engaging with the various elements of your immersive exhibition."* (J.Willson, Art Gallery of St. Albert, personal communication 2019)

There were 28 entries in the Art Gallery of St. Albert guestbook (Appendix I), and several of these corroborated the curator's comments, and reflected the viewers' engagement with both Green Point and the underlying concepts, in particular with the

concept of time as embodied in the work (Table 2)

### 5 Discussion

**5.1 Divisions over time: Connecting colonial world-views with the history of geology**

An impulse inherent in scientific exploration focuses on limits and boundaries of various types. The understanding of the extents of objects, natural phenomena, and concepts, their relationships, and the processes involved in their formation, rests

on human definition: a process that both includes and excludes (Bachelard, 1994, p.211–218). Thus, inclusion and exclusion rest as two faces, separated by a permeable and ever-shifting skin. Each category (and what it contains) fulfills specific needs at a given time. Borders or boundaries can also be subject to influences beyond their creators' control; they can take on a life of their own, invested with meaning and power beyond their initial scope (Nail, 2016).

The notion of a geological understanding of the land in 'deep time' (McPhee, 1981) and its implication and complicity with

colonial structures (Vance, 2017) comes into play here too. As Mohit Chanda points out,

*"the colonial project …defined the world as an extension of European frontiers…"*

and,

*"these colonially-generated spatial paradigms limit the definition of the world to its physical expanse, reducing all markers of plurality to a conquerable unit of spatial territory* (Chandna, 2009).

Geology – as a 'new' science – had a vital part to play throughout the exploration and colonization of Canada and Ktaqamkuk, now known as Newfoundland (Zeller, 2000). As a field of exploration and discovery, the study of the Earth provided (literally) valuable insights into resources available for use and development in newly-settled territories and for export back to home countries in Europe. As the history of exploration and settlement developed in Newfoundland, human relationships to the land shifted and evolved, including some and excluding others. The Indigenous Beothuk people carved new territory for themselves

inland from the coast to avoid contact with Europeans, but were killed by settlers, and succumbed to malnutrition and to diseases brought from Europe (Marshall, 2012; Rowe, 1977). The Mi'kmaq came seasonally to the west coast of Ktaqamkuk (Matthews and Robinson, 2018), to fish and hunt, and eventually settled in many areas (Bartels and Janzen, 1990; Martijn, 2003). Successive waves of European explorers and settlers came to the island – Newfoundland to the British, Terre Neuve to the French – to exploit its resources. At the present day, fishers maintain shoreline cabins a few hundred metres south of Green

Point at Green Cove. Many of the poles used in the construction of Boundary|Time|Surface bore traces of prior use in the construction of wharves, boat ramps, fish flakes and other structures used for fishing. The creation of Gros Morne National





Park and the designation of the Cambrian–Ordovician boundary stratotype on the island's west coast are just two more recent filters, with their associated boundaries, through which this coastal landscape can be viewed.

This palimpsest of histories informs both past and current views of the Green Point area. Despite the several-centuries duration
of human interaction with this shore, our inability to truly comprehend the vast amount of time represented in the cliffs throws into high relief both our insignificance in relation to the planet's long evolution, and simultaneously, our tremendous responsibility for our impact as a species in our brief existence on its surface (Singh, 2018; Wood, 2019). We have the option (and the choice) to reduce this impact: exploring the human relationship to geological "deep" time can be the basis for reevaluating what kind of animals we are, our relationship to the Earth. As actors on the geological stage, humans' erstwhile
convenient division between 'human' and 'natural' events is no longer relevant (Wood, 2019).

### 5.2 Global distribution of stratotypes

The Europe-centred development of geological science is well illustrated by a map of the worldwide distribution of type areas and stratotypes (Fig. 1). The original 19th century sites where periods of the geological time scale were defined are heavily concentrated in Europe and immediately adjoining areas. Many of the boundaries between these periods were subsequently
redefined by the ICS at GSSPs. These sites function as reference points, with which other places on the planet are correlated, so that Earth scientists can better understand whether changes that took place in the distant past were local or global in scope. At first glance, this process seems relatively straightforward: a point is chosen based on a set of criteria, and the boundary is set. But this is not the case. If the GSSPs were chosen purely on the basis of features intrinsic to the rocks – the excellence of the outcrop and its potential for correlation - an even distribution of GSSPs over the land surfaces of the Earth might be
expected. The actual distribution, though more dispersed than that of 19th century type-areas, still shows a strong bias toward European locations. The reasons for this become apparent when the arguments for the establishment of GSSPs are examined (Gradstein et al., 2012). In many cases, the final choice of a GSSP was made between fiercely contested candidates, each supported by a national scientific community centred in a political territory. Thus, a combination of objective and subjective influences came into play in determining the locations of these boundaries: the weight of evidence, interpretation of
information, *and* socio-political influences contributed to each decision. Thus Green Point was one of several places that could have been chosen for this particular boundary stratotype. As such, this place embodies the nexus of many aspects of the human pursuit of knowledge – and the selectivity with which that knowledge is related and used. *Boundary|Time|Surface* illustrated both the power and the (potential) futility of the human impulse to divide up the world in various ways. This impulse to define, name, and contain – so evident in the scientific discourse around this particular place – can be correlated with (and often
utilized by) socio-political discourses that have shaped nations, our understanding of who we are, and where we belong.

### 6 Conclusions

Gros Morne National Park – and Green Point in particular – lends itself perfectly to integrating artistic expression with scientific understanding of the natural world. The locale afforded the opportunity not only to create a large sculptural installation with immediate visual and metaphorical impact, but also to make work that blurred the boundaries of artistic and
scientific practice in a tangible way. Visitors to the site were able to engage with both the scientific and artistic aims of the project on a number of levels simultaneously, as they had a tangible, visual 'anchor' for the underlying ideas. Extending the reach of the original installation through public talks, the development of gallery presentations of new work, and the publication of a book has allowed *Boundary|Time|Surface* to be experienced in a number of different ways by much larger numbers of people since its initial creation. Using a range of strategies to convey both scientific and socio-political concepts associated
with the original ephemeral installation has provided multiple entry points for a wider audience to appreciate the geology and



history of Green Point, and geology as a human endeavour. Further, the temporal quality of work derived from the original installation invites viewers to consider the different scales of time present in the original site, and by extension, provides an opportunity to contemplate the human concept of time in relation to our actions on the planet vis á vis the scale of geological time. Both the original installation and the work developed subsequent to that project allow audiences to explore the ways in

which humans understood and acquired knowledge about this place, and how a particular world-view always informs a process of inquiry (Bachelard, 1994, p.212), even if it remains unacknowledged. Further, the exhibition environment, in particular, offered the opportunity for contemplative reflection, allowing viewers the physical and mental space to consider their own assumptions and those of others in relation to time and their role on the planet. The original work and the various methods of communicating the experience of its brief existence is an ongoing project to destabilize the fantasy that humans are somehow

separate from the Earth (Boetzkes, 2010, p.18), its systems and timescale – and the notion that borders, boundaries, and other forms of territoriality are somehow permanent.

**Author contribution.**

SAL created the artistic content and wrote the sections of the text describing these aspects. JWFW wrote the sections on stratigraphy and history of geology. Both authors collaborated in the editing and diagram preparation.

**Acknowledgements**

The authors acknowledge Parks Canada and the Art in the Park program for the residency at Gros Morne National Park, and Rob Hingston, Munju Ravindra, Kirsten Oravec, Fred Sheppard many other Parks Canada Staff; financial support from the Edmonton Arts Council and the Alberta Foundation for the Arts; and the efforts and enthusiasm of volunteers Michael Burzynski, Ryan Lacombe, Lisa Liu, Anne Marceau, Renée Martin, and Shawna White, who assisted in the construction of

the installation at Green Point.

**Competing interests**

As a professional artist, SAL has an interest in the sale of works derived from the project described in this paper.

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



**Figure captions**

**Figure 1. (a) Global location of 19th century definition areas of systems in the geological timescale, compared with boundary stratotypes defined and proposed by the International Commission on Stratigraphy (ICS) (Gradstein et al., 2012). Box encloses area of Fig 2(a). Molleweide projection; ICS colour scheme for stratigraphic units. (b) Enlarged portion of (a) showing concentration of definition areas and stratotypes in Europe.**

**Figure 2. (a) Main tectonic subdivisions of Newfoundland, showing location of Green Point stratotype (Lacombe et al., 2019). (b) Satellite view of Green Point stratotype area, showing location of the installation and recording locations. Imagery copyright 2020 CNES/Airbus, Landsat/Copernicus, Maxar Technologies, Map data copyright 2020 Canada.**

**Figure 3. Completed installation viewed at low tide. (a) Completed Boundary|Time|Surface installation, artist for scale. (b) View of completed Boundary|Time|Surface installation with viewers engaging with the work.**

**Figure 4. Samples from time-lapse sequence of photographs taken at 10 s intervals during construction and dissolution of the installation. (a) Site immediately before installation began. (b)Start of installation on falling tide. (c) Towards end of installation. (d) Completed installation near low tide. (e-h) Afternoon and evening dissolution of the sculpture during rising tide.**

**Figure 5. (a-d) Still images from GoPro™ first author's head-mounted camera taken during installation process.**

**Figure 6. (a, b) Examples of pencil slate sculptures built on the Cambrian–Ordovician boundary after the main Boundary|Time|Surface installation. (c) Construction of sculptures. Artist for scale.**

**Figure 7. a. Introductory panel, and b, one of two didactic panels created by the authors, using their own work and images in the public domain, for use in Discovery Centre Exhibition. Larger versions and sources are provided in the supplement. Additional imagery and French translation were provided by Parks Canada.**

**Figure 8. Mounted exhibition in gallery setting. (a) Discovery Centre Gallery, exhibition view showing projection-mapped video installation with driftwood logs and beach cobbles in background. (b) Art Gallery of St. Albert, exhibition view showing silk organza panels, video installation, and print works in the background. (c) View of photo-based print installation showing levels of transparency in the work. (d) "The Historic Coast" - multi-layer gel-transfer print work showing topographical map, enlarged seismic profile, photo of green point cliff, satellite image of green point, historical book cover, images of conodont teeth. 91 cm x 91 cm x 13 cm. (e) "167 Lifetimes" - gel-transfer print work showing enlarged detail of bedded limestone and shale in outcrop, Green Point NL; glass paint used to create 167 tick marks across image, each representing one 80-year human lifespan. 91 cm x 91 cm x 4 cm.**



**Table 1: Presentations by Date and Location**

| Date<br>dd-mm-year | Location | Approximate Audience | Audience Type |
|---|---|---|---|
| 13-06-2014 | Galliot Studios, Woody Point NL – Artist's Talk | 20 | General |
| 31-10-2015 | ATLAS Speaker's Series, University of Alberta, Edmonton AB | 30 | Scientific/Academic/Student |
| 31-01-015 | Education and Outreach Session, Atlantic Geoscience Society Colloquium, Truro NS | 20 | Scientific/Academic |
| 27-03-2015 | Edmonton Geological Society Banquet, Edmonton AB | 45 | Scientific/Academic |
| 15-07-2015 | Fundy Geological Museum, Parrsboro NS | 20 | General |
| 21-05-2016 | Gros Morne Discovery Centre, Woody Point, NL | 20 | General |
| 23-11-2016 | Fine Art Speaker's Series, MacEwan University, Edmonton AB – Artist's Talk | 25 | Academic/Student |
| 28-02-2019 | LaserAlberta Speaker's Series, University of Alberta Department of Art & Design, Edmonton AB. | 50 | Academic/Student/General |
| 14-09-2019 | Art Gallery of St. Albert, AB – Artist's Talk | 35 | General |
| 05-11-2019 | Acadia University, Wolfville NS. | 20 | Scientific/Academic/Student |
| 29-11-2019 | Earth Sciences Speaker's Series, McGill University, QC | 25 | Scientific/Academic/Student |





535 **Table 2. Examples of visitor responses to exhibitions at the Discovery Centre, Gros Morne National Park, NL, Canada, and the Art Gallery of St. Albert, AB, Canada. Complete visitor responses are listed in the supplement.**

| **Discovery Centre, Gros Morne National Park, NL Canada, 2016** |
|---|
| "TIME – the time, how it is stretched and/or tightened during my engagement … " |
| "Simplistic, yet effective. Why do we set up boundaries? Why is space divided and not opened up to shared use?" |
| "Fascinating & I'd love to Visit Green Point. Wonderful concept." |
| "I felt the definition of time with this work. Space to breathe, moments of stillness while surrounded by natural movement and progression of time." |
| "Beautiful work, love the layers and sense of time. Thank You." |
| "LOVE GREEN POINT!!!" |
| "Inspiring Geography – inspiring art." |

| **Art Gallery of St. Albert, AB, Canada, 2019** |
|---|
| "All of my senses are smiling." |
| "Love the layering." |
| "Good historical data." |
| "Good exploration of the concept of boundaries." |
| "Amazing and profound - my father is a geologist and I will tell him about this." |
| "Thought provoking – If there were no boundaries in the world, perhaps there would be less problems." |





fig 01

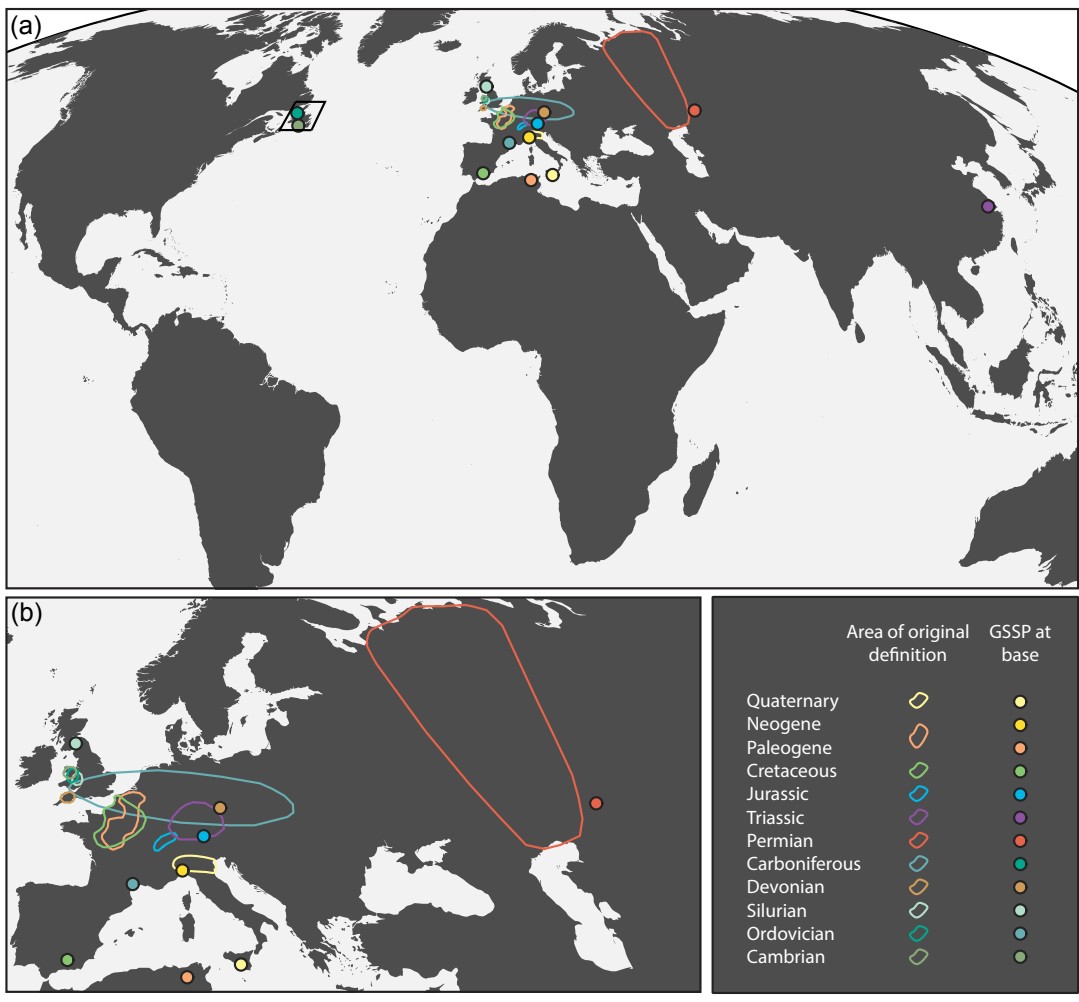



fig 02





fig03

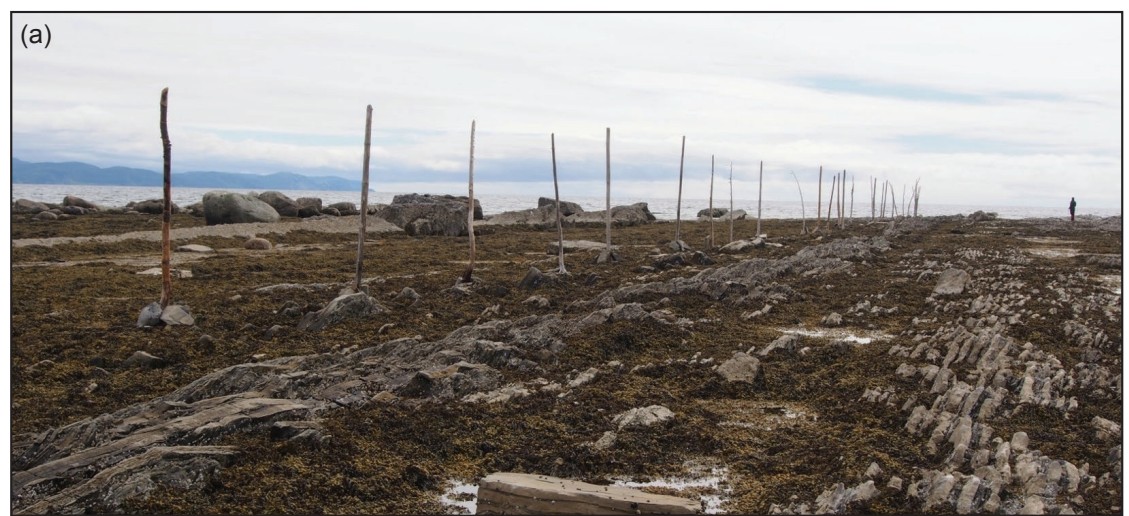

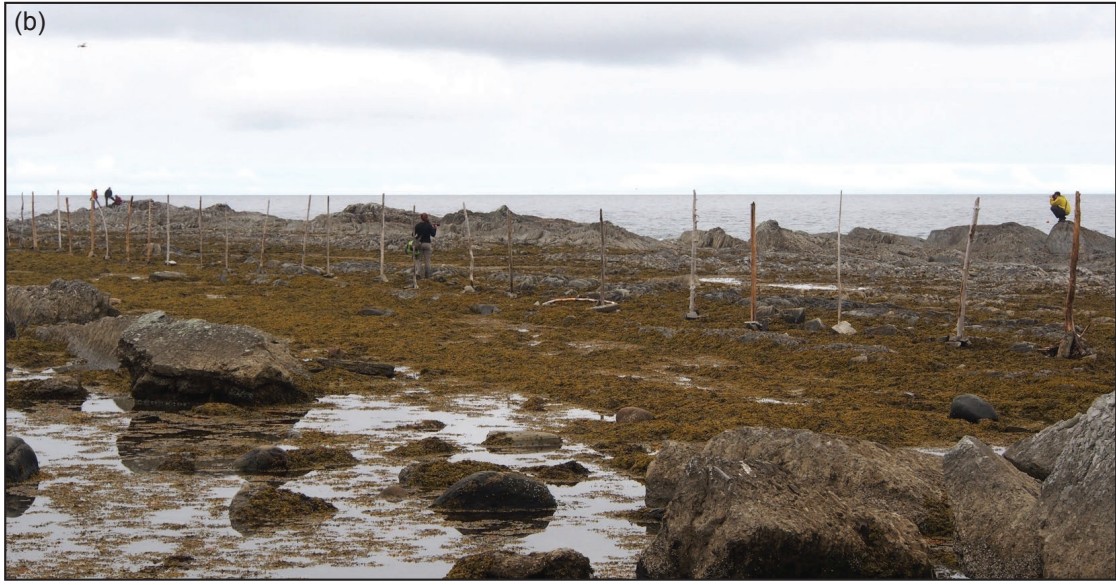



fig 04





fig 05

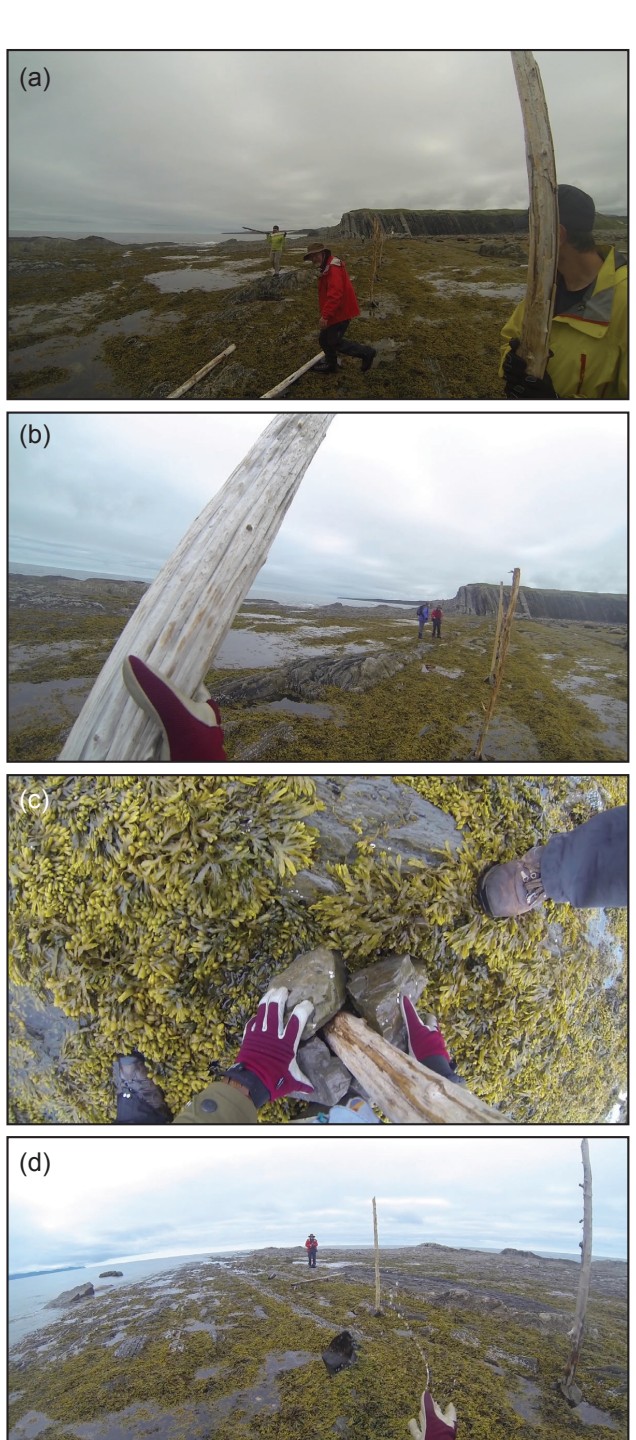



fig 06

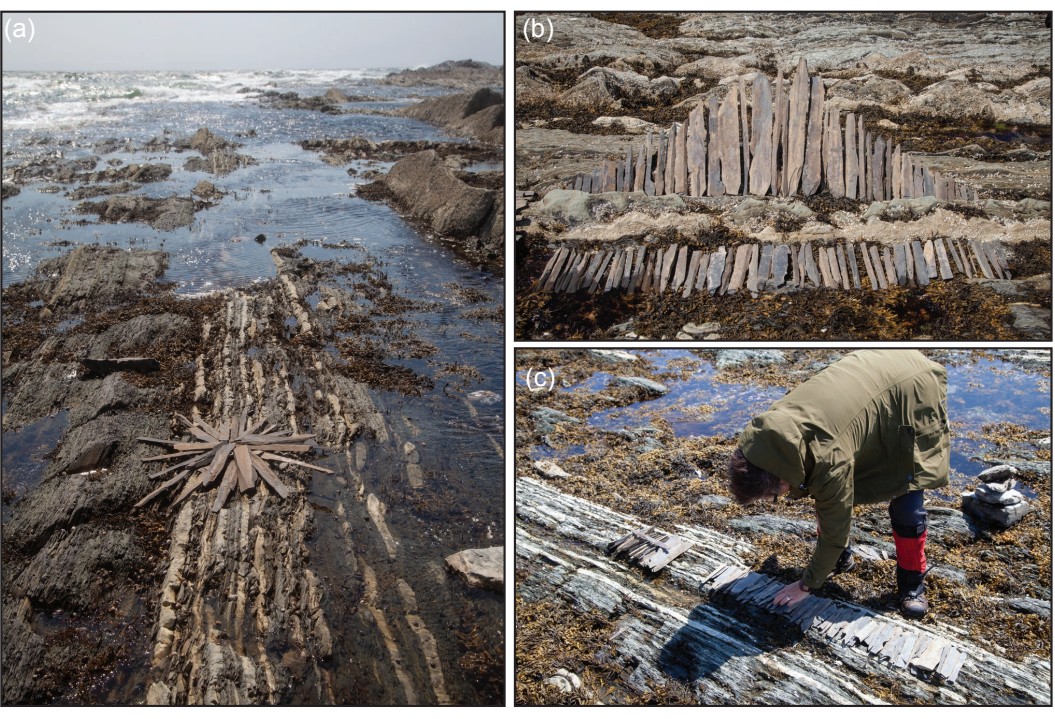



fig07

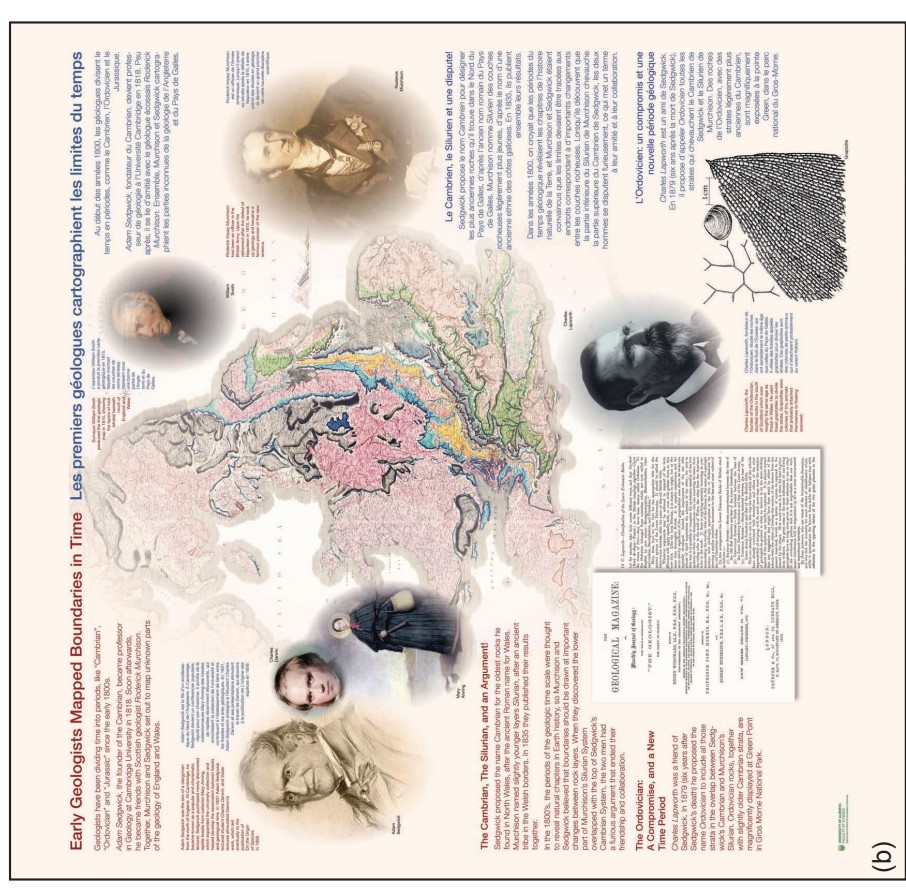

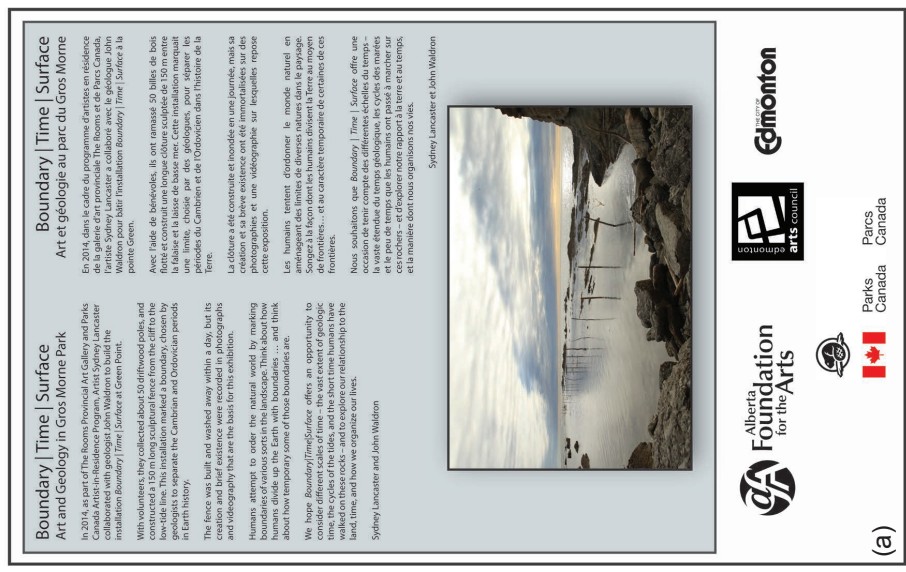





fig08

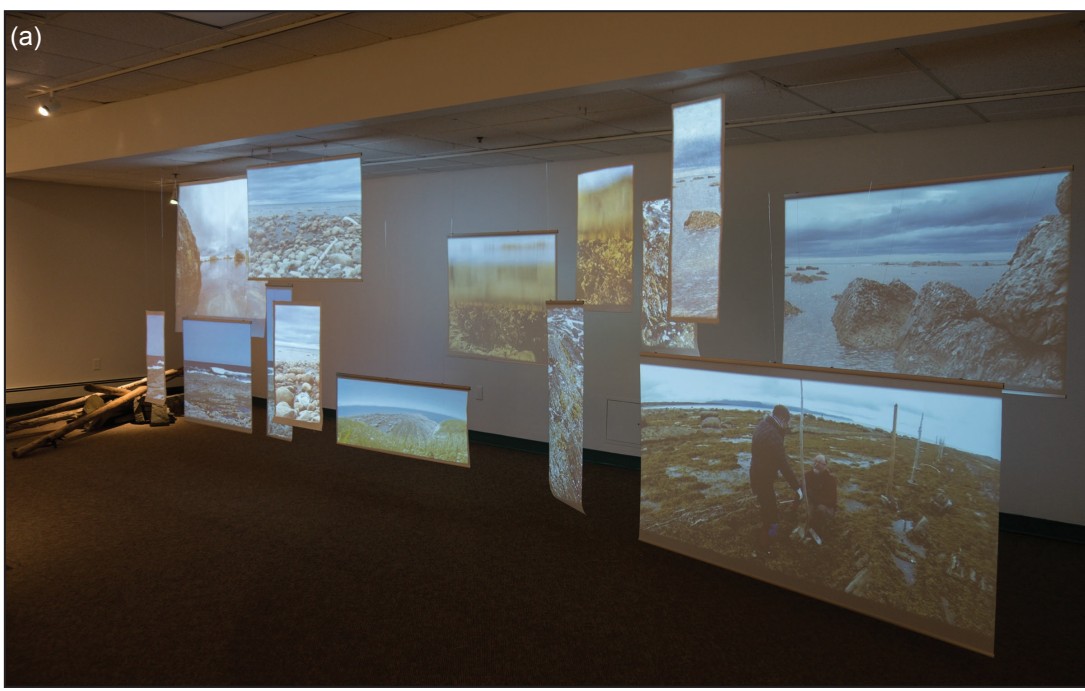

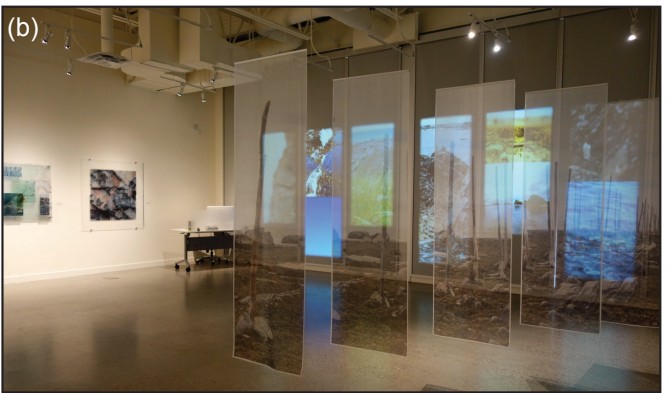

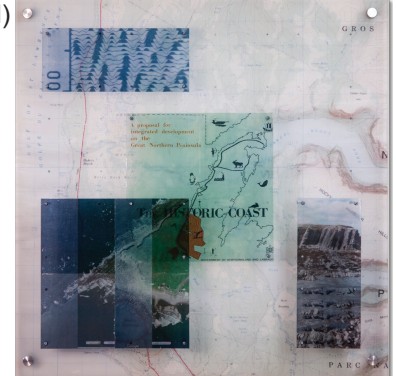

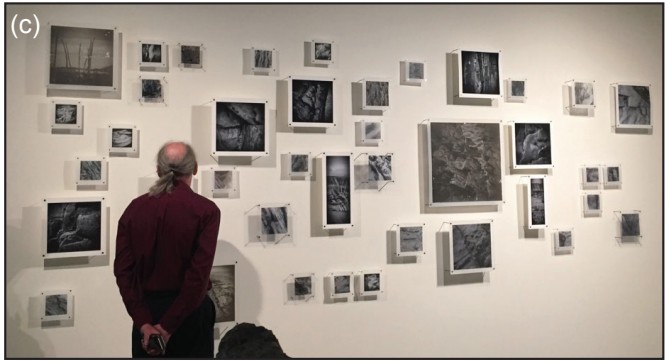

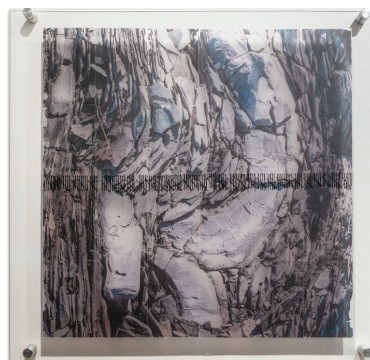