# Peer review of "Boundary|Time|Surface: Assessing a meeting of art and geology through an ephemeral sculptural work Sydney A. Lancaster1, John W.F. Waldron2"

_Geoscience Communication, 2020_

## Referee Comment (RC1) · Tim Fedak (Referee) · 3 Apr 2020

In their discussion paper, Lancaster and Waldron provide a detailed summary of the intention, planning, and results of a geoscience art project undertaken at Gros Morne National Park. The Boundary|Time|Surface project connects the history of geoscience and the stratigraphic boundary of the Silurian with an ephemeral site-specific art installation. Situating this art-science collaborative project within a history of earthwork art practice, the paper provides details of how the public engaged with the art during installation and up to five years after the earthwork sculpture had been washed away by the tides. The paper concludes with thoughtful reflections on conceptual implications

and boundaries in definitions of self and the earth. It is this type of project, blurring the boundaries between art and science – where new concepts and insights remain to be discovered in the future.

It turns out that I experienced one of the public lectures about this project, so have the benefit of offering a perspective as a participant. During the public presentation in July of 2015, the authors explained their project to an audience of twenty in a small coastal community of Parrsboro. Context of place is important, of course. Parrsboro was where Abraham Gesner was a physician when he examined and mapped the area's geology in the early 1830s, and also the land called Mi'kmawey by the Mi'kmaq people who have used the geology of the area for thousands of years. Communities that exist within geologically significant landscapes develop a profound appreciation of the culture of geology. The audience at the museum was indeed inspired by the project, and as a result other collaborations developed with local art groups.

The documentation of an ephemeral artwork provides an opportunity for expanding the interpretations, meanings, and impacts across different communities. As we all continue to rapidly adapt to new worldviews, most recently of an increasing concern for climate change and an increased awareness of global connectivity as we all adjust to a pandemic, boundaries, time, and space, continue to change in order to respond with compassion and a collective good. We see the world differently through a lens of climate and geology.

The history of stratigraphy, boundaries, and early geological maps, open our minds to the fundamental questions that challenged early geoscientists and their results that ushered in new global worldviews and discoveries. The age of the earth, plate tectonics, global processes, and our first steps on the moon can all be traced back to the first geology maps that began to document, delineate, and define the observed geology structures from which new discoveries would be made today. Marks on a map, signifying a place, in space and time.

Today we now live in a hyper-digital interconnected culture and expanding worldview. Although an ephemeral earthwork installation, the Boundary|Time|Surface project, those 52 vertical driftwood poles aligned like beacons on a beach, continue to be considered, discussed and to inspire thoughts about time, and our place on this earth. Perhaps there are no real boundaries between man vs nature. Perhaps we are, after all – simply a part of this earth; the earth looking at itself and wondering about the vastness of our geological history.

In preparing this review of Lancaster and Waldron's discussion paper, my hope is that their discussions, collaborations and discoveries will continue. With this publication, the 52 poles remain standing; as a memory, for future discovery. What new audiences will engage with these ideas, and in what new ways? We will see.

---

## Short Comment (SC1) · 10 Apr 2020

During a recent virtual writing retreat, we used a peer-review framework to review your abstract. We then had an open discussion and noted down all the feedback and compiled the following. We reviewed your Abstract using a structured worksheet with the following advice in mind: "The abstract is a condensed and concentrated version of the full text of the research manuscript. It should be sufficiently representative of the paper if read as a stand-alone document". We looked for important elements of a research abstract and we comment on them below. We hope the following is helpful for your revisions.

[Figure]

It's important to note that Geoscience Communication puts a lot of emphasis on evaluation of communication practice and ensuring that the practice is based on a solid foundation and research question. The articles need to tell the story of research on geoscience communication and not just tell the story of geoscience communication that's been done.

Overall: The project sounds very interesting and it's great when artists and scientists work together like this. The Abstract excited many of us, and we wish we would have seen the exhibition in real-life. The Abstract touches on some very interesting elements and issues, which made us want to read on. However, there are a few things we believe should be improved for this to be relevant for a research journal like Geoscience Communication.

Title: The title matches the abstract, but not necessarily the objective of the study, which we did not manage to grasp. It's a clear title, but we wondered if the location needs to be mentioned. Why not simply write "geological boundary"? Also, if you include the location in the title, then this should match how you describe the location in the Abstract itself. At the moment, they do not seem to match, which makes it difficult for readers not familiar with these places.

Need and relevance: We failed to clearly identify what the need or relevance of this work from the Abstract. If it is to "interrogate the human practice of dividing the Earth for social, political, scientific and aesthetic reasons" then could you add a short sentence to explain why. We also didn't quite understand what these "social, political, scientific and aesthetic reasons" might be. We have a feeling that the second and third paragraphs on touching the relevance of the project, but the ideas need to be drawn together. For example, we thought that the "dialogue with the land and environment" sentence in the third paragraph seems connected to the projects need and/or relevance, but we're unsure exactly how.

Hypothesis/Objectives: This is where we had difficulty. We can't see any research

objectives, questions or hypotheses. If the aim is to interrogate human practice, then what is the question related to that and how does one evaluate it?

Methods: The method was not clear probably because the research question was also absent. The authors included a description of the process itself including the construction and destruction of the artwork, and also the exhibitions, and book that came thereafter. This makes a nice story about the art itself, but lacks a method for the evaluation, which it needs to be a full story about research.

Results and conclusion: From our understanding, the results were the exhibition and book which were explained well and placed towards the end. However, this is not enough. The results of the research process need to explain the results related to the research question itself and the evaluation that's been carried out. We are left wondering about the impact of the project. Did it contribute to the collaboration between artists and scientists for example? Did it make people talk (or interrogate) the human practices mentioned in the beginning?

Take home message: A take-home message will ideally mention how the research contributes to a wider perspective. The last sentence should sum up the essence of the paper. At present the final sentence does not do this. We would have loved to see the resulting book, but we have to ask if a "limited" edition book helps to extend the reach of the project more than just in a "limited" way.

Clarity and conciseness: The abstract is mostly easy to understand. The authors should consider reformulating the second sentence of the second paragraph which is particularly difficult to follow. And maybe the authors would consider splitting or editing some of the longer sentences to shorter forms. Our main issue concerns the flow in the abstract which is quite challenging to follow. It is challenging to pick out the main elements of the paper. We hope the authors consider restructuring the Abstract in an order like this: Need/relevance, research question/hypothesis, methods, results, conclusions and take-home message. In this way the abstract should mirror your paper

and include all the important elements that your paper likely already contains.

Hopefully all the elements we felt are missing in the Abstract can easily be extracted from the paper itself. Since we were peer-reviewing the abstract as a stand-alone piece of writing, we did not investigate the rest of the paper, so we cannot comment on whether or not these elements are there.

A lot of our comments stem from the lack of a clear research question. Overall this seems like a very nice story about geoscience communication, and we hope that you can add the needed information to make it a nice story about research into geoscience communication.

Mathew Stiller-Reeve (Thematic Editor of Geoscience Communication) and 11 anonymous reviewers.

Please also note the supplement to this comment:
https://www.geosci-commun-discuss.net/gc-2020-2/gc-2020-2-SC1-supplement.pdf

---

## Referee Comment (RC2) · Graham Young (Referee) · 28 Apr 2020

This paper describes and illustrates what must have been a fascinating work of art to see, an ephemeral piece that lasted a very brief time. I find the artwork compelling, but I do have questions about the way in which the text is written, some of them perhaps related to the underlying philosophy, but also reflecting the difficulty I had in understanding how the art may relate to that philosophy.

I wonder why the authors state only that they intend to interrogate the human practice. Why not also look at the natural forces that created a situation where it was logical to place a system boundary? Overall, in their focus on humanity and on the interpreted

colonialism of pioneering geologists, they gloss over the extent to which the recognition of geological boundaries in the modern world is driven by the rocks themselves, and what they tell us about the changing world. The Green Point boundary is not the same boundary that was erected by 19th century British scientists; rather, it is the outcome of 20th century international negotiation, and of collaborative efforts by people from many countries to understand past global events. Even though the process of selection of global stratotypes is discussed near the end of the paper, it may be that the overall focus on colonialism and the imposition of human will on the world is not a particularly good fit for the particular boundary to which the art was applied.

Also, how does the work interrogate this human practice? Although this is brought up various times through the paper, it is not clear to me how the work really does this. Still, it is/was a beautiful and intriguing piece of landscape art.

The authors state (lines 388-391) that "The original work and the various methods of communicating the experience of its brief existence is an ongoing project to destabilize the fantasy that humans are somehow separate from the Earth (Boetzkes, 2010, p.18), its systems and timescale – and the notion that borders, boundaries, and other forms of territoriality are somehow permanent". This may be true for some systemic boundaries, since several of them are quite arbitrary, but there are others (the K-Pg is the most obvious example, but the O-S is another) that stand out "like a fish in a tree". The boundaries that are placed at mass extinction horizons are, indeed, permanent - they impose themselves on the viewer, rather than the other way around.

There is a lot of information on the technical aspects of constructing the artwork, but I wonder about other things the authors might have done in addition to the recording that was carried out. Did they consider virtual reality 360 photography? This would certainly have brought the record closer to the actual piece, reducing the suggestion that documentation is an 'edited version' of what once existed. If VR was considered, why wasn't it used?

Similarly, in this modern world, why was there not a web version? Gallery exhibitions, talks, and a book are all very traditional and "niche"; an online presence could have reached (or could still reach) a much larger, global audience. It might have also generated more audience engagement and response. The video installations seem as though they could have been very effective - I wish I could have seen them. Are there any thoughts about posting these to the web?

Specific Comments

Line 35-40 - What was the island called by the Beothuk? The Beothuk name, if known, should probably take precedence over all subsequent names.

2.1 Social and cultural context - This discusses the idea of boundaries as constructs of humanity, but ignores the fact that many boundaries are also natural features. In space, boundaries between countries or territories are often rivers or coastlines. In geological time, boundaries are often placed at very distinct geological event horizons - the Cretaceous-Paleogene boundary would be the best-known example of this.

Line 140 - The way this is written, it suggests that the Silurian-Devonian boundary is still at the Ludlow Bone Bed - maybe add a word such as "initially" or "originally" to "placed at the Ludlow Bone Bed".

Line 204 - What equipment was used for photography and videography?

Line 345-350 - In discussing our inability to truly comprehend the vast extent of geological time, it might be useful for the authors to circle back to the role of boundaries in understanding this time. Any glimmer of understanding that we now possess is largely the outcome of that exercise of defining geological periods and the boundaries between them.

Technical Corrections

Abstract, Line 29-31 - wording reads as though the public are a range of visual media

Line 72 - Add space between "thus" and "far"

Line 87 - Try to rephrase, to tidy up usage of "which" and "that" - refer to a style guide for appropriate usage.

Line 121 - Fix punctuation - at the moment it reads as though Sedgwick's father was elected to the Woodwardian chair.

Line 219 - Is "dissolution" the best word for what happened to the installation? It was destroyed (disarticulated and abraded), rather than dissolved.
* * *

---

## Referee Comment (RC3) · Simone Rödder (Referee) · 10 Jun 2020

The paper presents an outdoor ephemeral art installation situated at the intersection of land art and the discipline of geology. While not a research paper in a narrow sense, the paper outlines the rationale for the artwork, presents its science as well as arts context, describes its methodology and set-up, the installation process and results in the form of lasting documentations as well as audience reception. In my view, the title in its current form captures this overall scope well and should not be changed to include one of the sub-points mentioned above.

With its idea to add a scientific publication to the range of lasting documentations of the

artwork (this could be made more explicit in the paper as another goal), the manuscript in my view is pertinent to the journal and deserves the attention of the geoscience community.

From my point of view as a social scientist, however, the paper in its current form strongly reflects the background of its author team, an artist and a geoscientist. The chapter on the history of the C-O boundary (2.3.) serves to motivate the selection of the installation site, yet it is too long and detailed for requirements. In addition, some of the jargon should be explained (e.g., stratotype) or removed (e.g., stratigraphic codes) to make the paper accessible to a broad audience.

In a background chapter, the artwork presented in the paper is situated in an arts context. What the paper lacks, however, is a more systematic conceptualization of the social context of the project (2.1), and namely the art-science collaboration. In the social sciences, the term "boundary" is regularly used to refer to communication between professional worlds such as arts and science. Art-science collaborations then can be understood themselves as communication across boundaries including boundary work and boundary objects (e.g., Rödder 2017,"The climate of science-art and the art-science of the climate: Meeting points, boundary objects and boundary work ". Minerva, 55). I think the paper would greatly benefit (& could address some of the other referees' concerns) by a more theoretical grounding of its concept of art-science, including a clearer and more explicit reflection on the many notions of boundaries that 'surface' in the paper: manifest, metaphorical-symbolic, permanent, positive (fixed points, security), negative (separation, exclusion).

l. 293 I think the economic interest should not be mentioned explicitly in a scientific paper.

---

## Author Response (AR1)

Thank you for your helpful comments regarding the abstract. We suggest some revisions we might make below.

Title: The title matches the abstract, but not necessarily the objective of the study, which we did not manage to grasp.

Response: We suggest the modifying the title to make the objective clearer: "Boundary|Time|Surface: Assessing the public response to a geologically themed art project in Gros Morne National Park, Canada"

It's a clear title, but we wondered if the location needs to be mentioned. Why not simply write "geological boundary"? Also, if you include the location in the title, then this should match how you describe the location in the Abstract itself. At the moment, they do not seem to match, which makes it difficult for readers not familiar with these places.

Response: In the regional geology literature, it is usual that editors require some definition of the location in the title. Also, the art is a site-specific installation; we therefore feel mention of the location is appropriate to the title, but will shorten it and edit the abstract so that the location descriptions in the title and the abstract match.

Need and relevance: We failed to clearly identify what the need or relevance of this work from the Abstract. If it is to "interrogate the human practice of dividing the Earth for social, political, scientific and aesthetic reasons" then could you add a short sentence to explain why.

Response: We will add sentences to the first paragraph: "One such practice is the subdivision of geologic time. We assess the role of this site-specific art installation and its documentation in drawing the attention of a broader public to a boundary of importance in this endeavour." We will also edit the second and third paragraphs (see below) to bring our their relationship to need and relevance, which were insufficiently clear.

We also didn't quite understand what these "social, political, scientific and aesthetic reasons" might be.

Response: We do feel that this summary should be ok, as fully defining these common fields of inquiry would require far more space than an abstract.

We have a feeling that the second and third paragraphs on touching the relevance of the project, but the ideas need to be drawn together. . ..

Response: We hope that the added sentences will clarify this relevance.

Hypothesis/Objectives: This is where we had difficulty. We can't see any research objectives, questions or hypotheses. If the aim is to interrogate human practice, then what is the question related to that and how does one evaluate it?

Response: The added sentence in the first paragraph beginning "We assess..." will now address this.

Methods: The method was not clear... to be a full story about research.

Response: We will add a clear statement of the methods at the end of the first paragraph: "It was brought to the public through exhibitions, public talks, and a book. To evaluate the success of this project, we examine the public responses to these activities through attendance records and written visitor comments."

Results and conclusion: ... results of the research process need to explain the results related to the research question itself and the evaluation that's been carried out. Take home message: A take-home message will ideally mention how the research contributes to a wider perspective.. ...

Response: Thank you for pointing out these omissions; this is covered in the paper but was inadequately represented in the abstract. We will add a paragraph summarizing the outcomes and take-home message along the following lines: Questions at 11 public presentations indicated a high level of engagement from both artists and scientists. Of several thousand visitors to exhibitions, 418 written comments reflected the viewers' engagement with both Green Point and the underlying concepts. Both the original installation and the subsequent work allowed audiences to explore the ways in which humans understand and acquire knowledge about the Earth, and how world-views inform the process of scientific inquiry.

Clarity and conciseness: The abstract is mostly easy to understand. The authors should consider reformulating the second sentence of the second paragraph which is particularly difficult to follow. And maybe the authors would consider splitting or editing

some of the longer sentences to shorter forms.

Response: We will combine the second and third paragraphs, shortening some of the sentences, and re-wording so as to highlight the need and relevance of combining artistic and scientific approaches to the Earth. Our draft reads as follows: "Geologists and artists have taken different approaches in documenting features of the Earth, and have communicated these approaches to largely different segments of the population. Geology has as its basis the establishment of limits and boundaries within the Earth. Pioneers of geology defined the periods of the geologic timescale with the intent of representing natural chapters in Earth history; from their colonialist perspective, it was anticipated that these would have global application. Since the mid-20th century, stratigraphers have attempted to resolve the resulting gaps and overlaps by establishing international stratotypes. Artists creating work in dialogue with the land and environment have taken a range of approaches, from major, permanent interventions to extremely ephemeral activities, some of which echo practices in geological fieldwork. Boundary|Time|Surface attempted to bring a combination of scientific and artistic discourse to a larger public. The installation was constructed by hand in one day, on the falling tide from materials found on site, in order to have minimal environmental impact. During the remainder of the tidal cycle, and those following, the fence was dismantled by wave and tidal action. This cycle of construction and destruction was documented in video and with time-lapse still photography."

Our main issue concerns the flow in the abstract... In this way the abstract should mirror your paper and include all the important elements that your paper likely already contains.

Response: With the modifications described above, the flow of the abstract now closely follows that of the paper.

...Overall this seems like a very nice story about geoscience communication, and we hope that you can add the needed information to make it a nice story about research

into geoscience communication.

Response: We thank the reviewers for their comments and hope that the modifications have addressed their questions.

Geosci. Commun. Discuss.,
https://doi.org/10.5194/gc-2020-2-AC2, 2020

[Figure]

Thank you for your helpful comments and review of this paper. We suggest some revisions we might make below.

I wonder why the authors state only that they intend to interrogate the human practice.

Response: We will add to the abstract (where this phrase occurs) a clearer indication that human practice includes scientific activity. We will also change the metaphorical "interrogate" to "draw attention to" as this is our more literal meaning.

Why not also look at the natural forces that created a situation where it was logical to

place a system boundary? Overall, in their focus on humanity and on the interpreted colonialism of pioneering geologists, they gloss over the extent to which the recognition of geological boundaries in the modern world is driven by the rocks themselves, and what they tell us about the changing world.

Response: we will add sentences to section 2.3.2 to give a more nuanced explanation of the different factors that have led to the location of boundaries, including the examples raised by the referee: the Cretaceous-Paleogene, Permian-Triassic, and Ordovician-Silurian boundaries, to highlight the combination of geological and pragmatic considerations that go into the placement of boundaries. Our proposed addition is as follows

'In some cases, such as the Cretaceous-Paleogene boundary, the traditionally identified horizon marks a sudden global change that is easily correlated worldwide. In other cases, such as the Permian-Triassic and Ordovician-Silurian boundaries, major global change occurs over an interval within which correlation is challenging. For pragmatic reasons, the Permian-Triassic boundary stratotype at Meishan, China, was therefore placed in the an interval with cosmopolitan fossils in the interval marking the first recovery from a major extinction event colloquially termed the "great dying" (Ord, 2012). Similarly, the Ordovician-Silurian boundary at Dob's Linn in Scotland is placed in black shales with abundant, well-described graptolites, somewhat above an interval of grey beds, lacking abundant graptolites, that records the "Hirnantian event" of global change to biotas (Cooper et al. 2012, Melchin et al. 2012).'

The Green Point boundary is not the same boundary that was erected by 19th century British scientists; rather, it is the outcome of 20th century international negotiation, and of collaborative efforts by people from many countries to understand past global events.

Response: We do feel that the text at the start of 2.3.3 makes this clear. Hopefully the additional context provided by the previous addition will help to bring this out.

Even though the process of selection of global stratotypes is discussed near the end of

the paper, it may be that the overall focus on colonialism and the imposition of human will on the world is not a particularly good fit for the particular boundary to which the art was applied.

Response: We believe that its choice is now better justified with the above additions to the text, which highlight the varying factors in play in the choices of stratotypes.

Also, how does the work interrogate this human practice? Although this is brought up various times through the paper, it is not clear to me how the work really does this. Still, it is/was a beautiful and intriguing piece of landscape art.

Response: We are glad that the reviewer found the installation both beautiful and intriguing, which was what we hoped would be the experience of viewers. We will modify the figurative "interrogate" to the more literal "draw attention to". In addition we will add several sentences within the text that highlight how this was achieved by the position of the structure within the relatively uniform succession of strata in the boundary interval.

The authors state (lines 388-391) that "The original work and the various methods of communicating the experience of its brief existence is an ongoing project to destabilize the fantasy that humans are somehow separate from the Earth (Boetzkes, 2010, p.18), its systems and timescale – and the notion that borders, boundaries, and other forms of territoriality are somehow permanent". This may be true for some systemic boundaries, since several of them are quite arbitrary, but there are others (the K-Pg is the most obvious example, but the O-S is another) that stand out "like a fish in a tree". The boundaries that are placed at mass extinction horizons are, indeed, permanent - they impose themselves on the viewer, rather than the other way around.

Response: This is certainly true for the K-Pg boundary, where the biological change is extreme and seems geologically instantaneous; we will adjust the text to provide the more nuanced perspective as suggested by the reviewer. We do note that at other boundaries that mark major global change, for example the O-S boundary mentioned by the author, and the P-Tr boundary that marks the end of the Paleozoic, the situation is not quite so clear. The boundary commissions have struggled with somewhat conflicting aims of marking the point that stands out by reason of most rapid change, and choosing a pragmatic stratotype that allows the most easy correlation to a well-studied and continuous section. Pragmatism has generally won out. For example, if memory serves, the peak of the Ordovician-Silurian Hirnantian event, interpreted as a major extinction driven by climate change, is typically marked in deep-water successions by grey beds in which graptolites are rare, in contrast to the black shales above and below. The eventual choice of stratotype, at Dob's Linn in Scotland, was slightly above the Hirnantian grey beds, in a section with better documented graptolites. Pragmatic considerations thus played a role in the final selection of the GSSP near to, but not at, an episode of global change. Similarly with the Cambrian-Ordovician boundary, the most obvious biosphere change was the advent of planktonic graptolites with Rhabdinopora flabelliformis. However, the boundary was eventually placed lower in the section between two very similar conodont species, because this provided better correlation. These factors are now discussed in added sentences at the end of section 2.3.2 (see above).

There is a lot of information on the technical aspects of constructing the artwork, but I wonder about other things the authors might have done in addition to the recording that was carried out. Did they consider virtual reality 360 photography? This would certainly have brought the record closer to the actual piece, reducing the suggestion that documentation is an 'edited version' of what once existed. If VR was considered, why wasn't it used?

Response: While we considered more advanced technology, neither VR nor $360°$ photography could be implemented due to financial constraints, including access to both equipment and expertise. (The residency at Gros Morne provided a basic stipend but no funding for equipment, some of which was borrowed from the Park.) The original project was undertaken in 2014, at which time these technologies were not as well developed or accessible as they are in 2020. From a philosophical and artistic point of

view we would contend that regardless of the method, documentation of site-specific work is unavoidably a 'translated' view of the original, as choices will be imposed both by the hardware and by the perspectives of those capturing and editing the images. More recent VR presentations we have seen in art exhibitions, while impressive, have not been the same as being in the real landscape; much of the artistic impact of VR stems from the degree to which they are selective representations of the real world.

Similarly, in this modern world, why was there not a web version? Gallery exhibitions, talks, and a book are all very traditional and "niche"; an online presence could have reached (or could still reach) a much larger, global audience. It might have also generated more audience engagement and response. The video installations seem as though they could have been very effective - I wish I could have seen them. Are there any thoughts about posting these to the web?

Response: Video documentation of the exhibitions has been shot, and could eventually be made available on the web. An important consideration is that working artists, without permanent positions in academia, galleries and museums, must retain some control over the distribution of their work in order to have any possibility of generating income. Galleries expect the material presented to be unique and not simultaneously available on the web. Nonetheless, samples of the graphical and video material are available at the web site of the first author; we had refrained from promoting these in the paper lest it be seen as an improper use of this medium.

Specific Comments Line 35-40 - What was the island called by the Beothuk? The Beothuk name, if known, should probably take precedence over all subsequent names.

Response: We agree completely; research we have conducted to date has provided no indication of what the Beothuk called the island. The extant wordlists date from the 19th century, and are records of common nouns, numbers, and the like recorded from some of the last living Beothuk. We would welcome input from others on this point.

2.1 Social and cultural context - This discusses the idea of boundaries as constructs

of humanity, but ignores the fact that many boundaries are also natural features. In space, boundaries between countries or territories are often rivers or coastlines. In geological time, boundaries are often placed at very distinct geological event horizons - the Cretaceous-Paleogene boundary would be the best-known example of this.

Response: While there are most certainly socio-political boundaries constructed along natural features, there is no absolute necessity for that to be the case in human relationships; it is a matter of pragmatism, negotiation and sometimes coercion. Likewise, these human-defined boundaries can be redefined repeatedly over time, as part of process of colonization and war (an excellent example can be found at https://www.youtube.com/watch?v=UY9P0QSxlnI).

Line 140 - The way this is written, it suggests that the Silurian-Devonian boundary is still at the Ludlow Bone Bed - maybe add a word such as "initially" or "originally" to "placed at the Ludlow Bone Bed".

Fixed.

Line 204 - What equipment was used for photography and videography?

Response: We will supply a complete list of all equipment used as an appendix to the paper.

Line 345-350 - In discussing our inability to truly comprehend the vast extent of geological time, it might be useful for the authors to circle back to the role of boundaries in understanding this time. Any glimmer of understanding that we now possess is largely the outcome of that exercise of defining geological periods and the boundaries between them.

Response: We will insert a reference to the exercise of dividing geological time in this discussion:

"We have the option (and the choice) to reduce this impact: exploring the human relationship to geological "deep" time, and the widely spaced markers we have placed

within it, can be the basis for reevaluating what kind of animals we are, our relationship to the Earth."

Technical Corrections Abstract, Line 29-31 - wording reads as though the public are a range of visual media

Fixed

Line 72 - Add space between "thus" and "far"

Done

Line 87 - Try to rephrase, to tidy up usage of "which" and "that" - refer to a style guide for appropriate usage.

Done

Line 121 - Fix punctuation - at the moment it reads as though Sedgwick's father was elected to the Woodwardian chair.

Fixed

Line 219 - Is "dissolution" the best word for what happened to the installation? It was destroyed (disarticulated and abraded), rather than dissolved.

Changed to destruction

With thanks, Sydney Lancaster & John W.F. Waldron
* * *
Geosci. Commun. Discuss.,
https://doi.org/10.5194/gc-2020-2-AC3, 2020

[Figure]

Thank you for your perceptive comments regarding this project. We feel strongly that pursuing genuine collaborations between artists and scientists is of tremendous benefit to both areas on inquiry, and can lead to new and exciting ways of communicating a wide range of information and ideas to broader populations.

With Thanks,

Sydney Lancaster & John W.F. Waldron
* * *
Geosci. Commun. Discuss.,
https://doi.org/10.5194/gc-2020-2-AC4, 2020

[Figure]

We will closely review section 2.3, with respect to trimming some detail and adding explanatory notes with respect to discipline-specific terms, in order to make the paper more accessible to a non-specialist audience.

In section 2.1, we will edit the text to incorporate a more systematic description of our approach to the Art-Science collaboration, with attention to our overall approach to communication between our professional realms. To this end, we are finding the paper

cited in the review helpful in framing our edits, as we do Kagan (2015), Madsen (2018), Rogers (2012), and Loveless (2019). Although not strictly related, Naill (2016), which we have cited in our paper, also offers some interesting insights with respect to theorizing borders and boundaries, albeit in a difference context. The editorial comments from Malina et al in LEONARDO, Vol. 51, No. 1,2018 (doi:10.1162/LEON_e_01555) are also thought-provoking.

With respect to Line 291: We assume the author is referring to line 294 (where we mention the availability of the book for purchase). We can rephrase to "available internationally". Our intent here was to speak to the socio-economic context of the first author as an independent visual artist, who cannot make this book available to the broader community as a freely-available, open source document, as it is a potential source of income. This is, in itself, a 'boundary' that differentiates the approach and reality of scientific and artistic work, a factor which is now covered elsewhere in these discussions.

References Noted Above:

Kagan, S. (2015). 'Artistic research and climate science: transdisciplinary learning and spaces of possibilities'. Journal of Science Communication 14(01)(2015)C07.

Loveless, Natalie. How to Make Art at the End of the World: A Manifesto for Research-Creation. Durham & London: Duke UP, 2019.

Madsen, Dorte. 'Epistemological or Political? Unpacking Ambiguities in the Field of Interdisciplinarity Studies.' Minerva (2018) 56: 453-477. https://doi.org/10.1007/s11024-018-9353-5

Nail, T.: Theory of the Border, Oxford UP, Oxford., 2016.

Rogers, Hannah Star. Practices of Art and Science. PhD Dissertation, Cornell University, 2012.

GC-2020-2
**List of Changes Made, as per Reviewer Comments**

SC-1, Mathew Stiller-Reeve (Abstract Review)

1. Revised Title: Boundary|Time|Surface: Assessing the meeting of art and geology in Gros Morne National Park, Newfoundland, Canada"

2. Sentence added to first paragraph.

3. Clear statements added regarding Objectives and Methods added to first and second paragraphs

4. Final paragraph added to summarize results, conclusion and take-home message.

5. Second and third paragraph were combined, and re-worded to improve clarity and conciseness.

RC-2, Graham Young

1. Wording changed in Abstract for clarity, and to make explicit the inclusion of scientific activity in the broader realm of 'human practice.'

2. Section 2.3.2 revised to provide more explanation of factors impacting the location of geological boundaries, including the examples raised by the referee: the Cretaceous-Paleogene, Permian-Triassic, and Ordovician-Silurian boundaries.

3. Beginning of section 2.3.3 has been revised to further clarify the distinction between 19th Century and 20th Century stratigraphic practice.

4. Section 2.3.2 edited to make clear the range of considerations in placement of boundaries between the Ordovician-Silurian and Cambrian-Ordovician.

5. Addressed queries regarding use of VR/360 degree video and online presentation of work in Author Response.

6. Addressed query regarding the name for Newfoundland in Beothuk (unknown, as it is a dead language and no record in extant lexicons exists).

7. Addressed query regarding the relationship between natural features and human-defined boundaries in Author Response.

8. Corrected wording regarding Ludlow Bone Bed.

9. Equipment list suppled as an appendix, and reference to the Appendix made at the end of Section 3.3.

10. Technical corrections noted by the Reviewer were completed.

RC-3, Simone Rödder
1. Section 2.3 was trimmed of extraneous detail, and discipline-specific terms were defined or otherwise clarified for a non-specialist audience.

2. Section 2.1 was edited and text added to address our approach to collaboration across disciplines in an explicit way, and pertinent references were added. Both the title and abstract have also been edited to reference this information, while acknowledging the suggestion of reviewer Stiller-Reeve for a more informative title.

3. Line 291-294 - explicit reference to the economic aspect of the work removed.

**Boundary|Time|Surface: Assessing a meeting of art and geology in Gros Morne National Park, Newfoundland, Canada**

Sydney A. Lancaster[1], John W.F. Waldron[2]

[1]Edmonton, Alberta, T6E1G6, Canada
[2]Earth and Atmospheric Sciences, University of Alberta, Edmonton, T6G2E3, Canada

Sydney A. Lancaster: ORCID: 0000-0002-5843-3947
John W.F. Waldron: ORCID: 0000-0002-1401-8848

*Correspondence to*: Sydney A. Lancaster (sydneylancaster@ualberta.net)

**Abstract.** *Boundary|Time|Surface* was an ephemeral sculptural work created to draw attention to the human practice of creating boundaries: dividing the Earth for social, political, scientific and aesthetic reasons. One such practice is the subdivision of geologic time for scientific purposes. We assess the role of this site-specific art installation and its documentation in drawing the attention of a broader public to a boundary of importance in this endeavour. The 150-metre-long work comprised a fence of 52 vertical driftwood poles, 2-3 m tall, positioned along an international boundary stratotype in Gros Morne National Park, Newfoundland, Canada, separating Ordovician from Cambrian strata. It was brought to the public through exhibitions, public talks, and a book. To evaluate the success of this project, we examine the public responses to these activities through attendance records and written visitor comments.

Geologists and artists have taken different approaches in documenting features of the Earth, and have communicated these approaches to largely different segments of the population. Geology has as its basis the establishment of limits and boundaries within the Earth. Pioneers of geology defined the periods of the geologic timescale with the intent of representing natural chapters in Earth history; from their colonialist perspective, it was anticipated that these would have global application. Since the mid-20th century, stratigraphers have attempted to resolve the resulting gaps and overlaps by establishing international stratotypes. Artists creating work in dialogue with the land and environment have taken a range of approaches, from major, permanent interventions to extremely ephemeral activities, some of which echo practices in geological fieldwork.

The site-specific installation was constructed by hand in one day, on the falling tide from materials found on site, in order to have minimal environmental impact. During the remainder of the tidal cycle, and those following, the fence was dismantled by wave and tidal action. This cycle of construction and destruction was documented in video and with time-lapse still photography.

[revised manuscript text omitted]

Similarly, undertaking this project itself represented a challenging of disciplinary boundaries, and discipline-specific modes of thinking. Both artist and scientist had a desire to come to the project as "equal but different" in their expertise, and in their approaches to the planning and execution of the original installation work and subsequent elements of the project. This positioning of our roles and respective disciplines was an active choice, the product of several extended discussions in which each participant explored their assumptions about the work within the other collaborator's discipline. As such, these discussions could be described as 'boundary-work' (Gieryn, 1995; Rödder, 2017). The artist sought information about the processes and context behind the development of the geologic time scale, the history of stratigraphic research on the west coast of Newfoundland, and artistic elements that exist in the documentation of Earth science in geological maps and other publications. The geologist learned about the various practices of creating art works within the landscape, from ephemeral to relatively permanent (at least on a human time scale), and their relationship to more conventional, gallery-based art traditions. These areas of knowledge are summarized in sections 2.2 and 2.3, below. We were concerned with establishing "common ground" both in terminology and in approach to the work to be undertaken, as the initial installation project was an

extended time commitment, and physically demanding. As such, we were self-selecting for co-production of the work at this stage (Rödder, 2017) and for the subsequent publication of an artist book derived from the original work. In that case, coming to agreement on the book's visual and written content, as well as its layout and design, was important to the success of that part of the project for both collaborators. Beyond these pragmatic considerations, however, we felt that our efforts could contribute to a process "whereby different modes of knowing, from outside science (or outside art), are engaged with" to offer a "wider integrative framework" (Kagan, 2015) and 
[revised manuscript text omitted]

645

---

## Editor Decision (ED1)

Dear authors,

I appreciate the originality of your work and relevance of the geoscience engagement.
Also, I acknowledge the revision of your manuscript following the comments of the reviewers.

However, before the paper can be accepted for publication in a scientific journal, there are some key issues still open. I kindly ask you for further effort to improve the presentation of your work and the readability of your manuscript following the standard rules of a scientific paper.

Title and abstract are the key presentation of your work. They need to be correctly drafted to help the reader to decide whether the rest of the paper is worth reading. Also, the Introduction, Background, and Discussion need to be reshaped to improve the flowing-sequence of information.
See also the annotated manuscript. I am giving you some hints there, but please, revise the whole manuscript in light of my comments.

1)      Title: I suggest  **Boundary|Time|Surface: Assessing a meeting of art and geology through an ephemeral sculptural work.**  It is now clear, at a first glance, what you have done. I  agree with the reviewer, here the location is not relevant.

2)      Abstract: Revise. It should provide a quick and accurate summary of the paper. Generally, it is one paragraph which summarizes the purpose, methods, results, and conclusions of the paper (100 -250 words is a good rule of thumb). You can still fulfill to the request of the reviewers using concise sentences. Some of the paragraphs now in the Abstract can be moved to the Introduction.

3)      Introduction. Here you should outline the problem and why it was worth tackling. Review the literature, recording briefly the main contributors and summarizing the status of the field when you started the research. State what you will do and what has not been done before. Keep it as brief as you can whilst still doing all this. You have provided much of this information in the Background section (sub. 2.1 and 2.2). Just need to organize better the contents and the sections.

4)      Move subsection 2.1 and 2.2 to Introductions. Summarize, list the facts and eliminate the anecdotal part.

5)      Section Background: reshape this section given the Introduction and avoid all the details that are not relevant to the work presented in this paper. This is valid throughout the paper. Make it clear the links between the contents of this section and the work of the exhibition.

6)      Change the title of Section 2:  **Geological background**; remove the title 2.3.1. (See the annotated manuscript)

7)      Pg. 5 - List the scientific facts and eliminate the anecdotal part.

8)      Change the title of section 3: **Boundary|Time|Surface: implementation of  the  exhibition**

9)      Change the title of section 4 – **Outreach and Communication activities**

10)      What about the "*Questions*" and the public engagement mentioned in the Abstract?

11)      Discussion. I found it difficult to follow this section. You should discuss your results and avoid waffling. Be clear and concise. Start from your results to extract principles, relationships, or generalizations (if any). Has the message that you wish to convey by the exhibition reached the general public ? List the strength and any reservations or limitations of your work here. There are some sparse considerations about your results in the other sections that should be moved here.

12)      References: In the text, it is enough the short citation of the papers- E.g. NO: (Gooding, 2002, p.21–23; The Art Story,) (Goldsworthy, 2000, p.74–77, 84–95, 122–129)  YES:  (Gooding, 2002); (Goldsworthy, 2000)
https://www.geoscience-communication.net/for_authors/manuscript_preparation.html

13)      Discuss all the figures presented in the paper. Fig. 3 is not mentioned

**Boundary|Time|Surface: Assessing a meeting of art and geology in Gros Morne National Park, Newfoundland, Canada**

Sydney A. Lancaster[1], John W.F. Waldron[2]

5   [1]Edmonton, Alberta, T6E1G6, Canada
[2]Earth and Atmospheric Sciences, University of Alberta, Edmonton, T6G2E3, Canada

Sydney A. Lancaster: ORCID: 0000-0002-5843-3947
John W.F. Waldron: ORCID: 0000-0002-1401-8848

*Correspondence to*: Sydney A. Lancaster (sydneylancaster@ualberta.net)

**Abstract.** *Boundary|Time|Surface* was an ephemeral sculptural work created to draw attention to the human practice of creating boundaries: dividing the Earth for social, political, scientific and aesthetic reasons. One such practice is the subdivision of geologic time for scientific purposes. We assess the role of this site-specific art installation and its documentation in drawing the attention of a broader public to a boundary of importance in this endeavour. The 150-metre-long work comprised a fence of 52 vertical driftwood poles, 2-3 m tall, positioned along an international boundary stratotype in Gros Morne National Park, Newfoundland, Canada, separating Ordovician from Cambrian strata. It was brought to the public through exhibitions, public talks, and a book. To evaluate the success of this project, we examine the public responses to these activities through attendance records and written visitor comments.

Geologists and artists have taken different approaches in documenting features of the Earth, and have communicated these approaches to largely different segments of the population. Geology has as its basis the establishment of limits and boundaries within the Earth. Pioneers of geology defined the periods of the geologic timescale with the intent of representing natural chapters in Earth history; from their colonialist perspective, it was anticipated that these would have global application. Since the mid-20th century, stratigraphers have attempted to resolve the resulting gaps and overlaps by establishing international stratotypes. Artists creating work in dialogue with the land and environment have taken a range of approaches, from major, permanent interventions to extremely ephemeral activities, some of which echo practices in geological fieldwork.

The site-specific installation was constructed by hand in one day, on the falling tide from materials found on site, in order to have minimal environmental impact. During the remainder of the tidal cycle, and those following, the fence was dismantled by wave and tidal action. This cycle of construction and destruction was documented in video and with time-lapse still photography.

[revised manuscript text omitted]

Similarly, undertaking this project itself represented a challenging of disciplinary boundaries, and discipline-specific modes of thinking. Both artist and scientist had a desire to come to the project as "equal but different" in their expertise, and in their approaches to the planning and execution of the original installation work and subsequent elements of the project. This positioning of our roles and respective disciplines was an active choice, the product of several extended discussions in which each participant explored their assumptions about the work within the other collaborator's discipline. As such, these discussions could be described as 'boundary-work' (Gieryn, 1995; Rödder, 2017). The artist sought information about the processes and context behind the development of the geologic time scale, the history of stratigraphic research on the west coast of Newfoundland, and artistic elements that exist in the documentation of Earth science in geological maps and other publications. The geologist learned about the various practices of creating art works within the landscape, from ephemeral to relatively permanent (at least on a human time scale), and their relationship to more conventional, gallery-based art traditions. These areas of knowledge are summarized in sections 2.2 and 2.3, below. We were concerned with establishing "common ground" both in terminology and in approach to the work to be undertaken, as the initial installation project was an

extended time commitment, and physically demanding. As such, we were self-selecting for co-production of the work at this stage (Rödder, 2017) and for the subsequent publication of an artist book derived from the original work. In that case, coming to agreement on the book's visual and written content, as well as its layout and design, was important to the success

90 of that part of the project for both collaborators. Beyond these pragmatic considerations, however, we felt that our efforts could contribute to a process "whereby different modes of knowing, from outside science (or outside art), are engaged with" to offer a "wider integrative framework" (Kagan, 2015) and 
[revised manuscript text omitted]

[Figure]

[Figure]

fig 04

[Figure]

fig 05

[Figure]

fig 06

[Figure]

fig07

[Figure]

[Figure]

fig08

[Figure]

(a)

[Figure]

(b)

[Figure]

(d)

[Figure]

(c)

[Figure]

(e)

---

## Author Response (AR2)

This section contains the editor comments, with responses embedded in the markup

Dear authors,

I appreciate the originality of your work and relevance of the geoscience engagement.
Also, I acknowledge the revision of your manuscript following the comments of the reviewers.

However, before the paper can be accepted for publication in a scientific journal, there are some key issues still open. I kindly ask you for further effort to improve the presentation of your work and the readability of your manuscript following the standard rules of a scientific paper.

Title and abstract are the key presentation of your work. They need to be correctly drafted to help the reader to decide whether the rest of the paper is worth reading. Also, the Introduction, Background, and Discussion need to be reshaped to improve the flowing-sequence of information.
See also the annotated manuscript. I am giving you some hints there, but please, revise the whole manuscript in light of my comments.

1)      Title: I suggest  **Boundary|Time|Surface: Assessing a meeting of art and geology through an ephemeral sculptural work.**  It is now clear, at a first glance, what you have done. I agree with the reviewer, here the location is not relevant.
[Figure]

2)      Abstract: Revise. It should provide a quick and accurate summary of the paper. Generally, it is one paragraph which summarizes the purpose, methods, results, and conclusions of the paper (100 -250 words is a good rule of thumb). You can still fulfill to the request of the reviewers using concise sentences. Some of the paragraphs now in the Abstract can be moved to the Introduction.

3)      Introduction. Here you should outline the problem and why it was worth tackling. Review the literature,
 recording briefly the main contributors and summarizing the status of the field when you started the research. State what you will do and what has not been done before. Keep it as brief as you can whilst still doing all this. You have provided much of this information in the Background section (sub. 2.1 and 2.2). Just need to organize better the contents and the sections.

4)      Move subsection 2.1 and 2.2 to Introductions. Summarize, list the facts and eliminate the anecdotal part.
[Figure]

5)      Section Background: reshape this section given the Introduction and avoid all the details that are not relevant to the work presented in this paper. This is valid throughout the paper. Make it clear the links between the contents of this section and the work of the exhibition.

6)      Change the title of Section 2:  **Geological background**; remove the title 2.3.1. (See the annotated manuscript)
[Figure]

7)      Pg. 5 - List the scientific facts and eliminate the anecdotal part.
[Figure]

8)      Change the title of section 3: **Boundary|Time|Surface: implementation of  the  exhibition**
[Figure]

9)      Change the title of section 4 – **Outreach and Communication activities**
[Figure]

10)     What about the "*Questions*" and the public engagement mentioned in the Abstract?

11)     Discussion. I found it difficult to follow this section. You should discuss your results and avoid waffling. Be clear and concise. Start from your results to extract principles, relationships, or generalizations (if any). Has the message that you wish to convey by the exhibition reached the general public ? List the strength and any reservations or limitations of your work here. There are some sparse considerations about your results in the other sections that should be moved here.

12)     References: In the text, it is enough the short citation of the papers- E.g. NO: (Gooding, 2002, p.21–23; The Art Story,) (Goldsworthy, 2000, p.74–77, 84–95, 122–129)  YES:  (Gooding, 2002); (Goldsworthy, 2000)
https://www.geoscience-communication.net/for_authors/manuscript_preparation.html

13)     Discuss all the figures presented in the paper. Fig. 3 is not mentioned

**Boundary|Time|Surface: Assessing a meeting of art and geology in Gros Morne National Park, Newfoundland, Canada**

Sydney A. Lancaster[1], John W.F. Waldron[2]

5    [1]Edmonton, Alberta, T6E1G6, Canada
[2]Earth and Atmospheric Sciences, University of Alberta, Edmonton, T6G2E3, Canada

Sydney A. Lancaster: ORCID: 0000-0002-5843-3947
John W.F. Waldron: ORCID: 0000-0002-1401-8848

*Correspondence to*: Sydney A. Lancaster (sydneylancaster@ualberta.net)

**Abstract.** *Boundary|Time|Surface* was an ephemeral sculptural work created to draw attention to the human practice of creating boundaries: dividing the Earth for social, political, scientific and aesthetic reasons. One such practice is the subdivision of geologic time for scientific purposes. We assess the role of this site-specific art installation and its documentation in drawing the attention of a broader public to a boundary of importance in this endeavour. The 150-metre-long work comprised a fence of 52 vertical driftwood poles, 2-3 m tall, positioned along an international boundary stratotype in Gros Morne National Park, Newfoundland, Canada, separating Ordovician from Cambrian strata. It was brought to the public through exhibitions, public talks, and a book. To evaluate the success of this project, we examine the public responses to these activities through attendance records and written visitor comments.

Geologists and artists have taken different approaches in documenting features of the Earth, and have communicated these approaches to largely different segments of the population. Geology has as its basis the establishment of limits and boundaries within the Earth. Pioneers of geology defined the periods of the geologic timescale with the intent of representing natural chapters in Earth history; from their colonialist perspective, it was anticipated that these would have global application. Since the mid-20th century, stratigraphers have attempted to resolve the resulting gaps and overlaps by establishing international stratotypes. Artists creating work in dialogue with the land and environment have taken a range of approaches, from major, permanent interventions to extremely ephemeral activities, some of which echo practices in geological fieldwork.

The site-specific installation was constructed by hand in one day, on the falling tide from materials found on site, in order to have minimal environmental impact. During the remainder of the tidal cycle, and those following, the fence was dismantled by wave and tidal action. This cycle of construction and destruction was documented in video and with time-lapse still photography.

[revised manuscript text omitted]

Similarly, undertaking this project itself represented a challenging of disciplinary boundaries, and discipline-specific modes of thinking. Both artist and scientist had a desire to come to the project as "equal but different" in their expertise, and in their approaches to the planning and execution of the original installation work and subsequent elements of the project. This positioning of our roles and respective disciplines was an active choice, the product of several extended discussions in which each participant explored their assumptions about the work within the other collaborator's discipline. As such, these discussions could be described as 'boundary-work' (Gieryn, 1995; Rödder, 2017). The artist sought information about the processes and context behind the development of the geologic time scale, the history of stratigraphic research on the west coast of Newfoundland, and artistic elements that exist in the documentation of Earth science in geological maps and other
 publications. The geologist learned about the various practices of creating art works within the landscape, from ephemeral to relatively permanent (at least on a human time scale), and their relationship to more conventional, gallery-based art traditions. These areas of knowledge are summarized in sections 2.2 and 2.3, below. We were concerned with establishing "common ground" both in terminology and in approach to the work to be undertaken, as the initial installation project was an

extended time commitment, and physically demanding. As such, we were self-selecting for co-production of the work at this stage (Rödder, 2017) and for the subsequent publication of an artist book derived from the original work. In that case, coming to agreement on the book's visual and written content, as well as its layout and design, was important to the success of that part of the project for both collaborators. Beyond these pragmatic considerations, however, we felt that our efforts could contribute to a process "whereby different modes of knowing, from outside science (or outside art), are engaged with" to offer a "wider integrative framework" (Kagan, 2015) and 
[revised manuscript text omitted]

[Figure]

(a)

[Figure]

(b)

fig 04

[Figure]

fig 05

[Figure]

fig 06

[Figure]

fig07

[Figure]

[Figure]

fig08

[Figure]

(a)

[Figure]

(b)

[Figure]

(d)

[Figure]

(c)

[Figure]

(e)

**Summary of Comments on Microsoft Word - 50 BTS article REVISED clean.docx**

**Page: 1**
* * *
Author: syd     Subject: Sticky Note   Date: 2020-07-19, 5:13:58 PM

We describe our changes in the comments below, and in responses to the individual flags on the pdf.
* * *
Author: syd     Subject: Sticky Note   Date: 2020-07-18, 3:24:19 PM

Done
* * *
Author: syd     Subject: Sticky Note   Date: 2020-07-19, 4:17:00 PM

Abstract revised to a single paragraph of 246 words
* * *
Author: syd     Subject: Sticky Note   Date: 2020-07-19, 5:13:09 PM

Introduction edited, incorporating the material formerly in subsections 2.1 and 2.2, shortened, and reorganized, as per editor's direction.
* * *
Author: syd     Subject: Sticky Note   Date: 2020-07-19, 4:10:46 PM

Done, following the editor's recommendations
* * *
Author: syd     Subject: Sticky Note   Date: 2020-07-18, 3:36:52 PM

DONE
* * *
Author: syd     Subject: Sticky Note   Date: 2020-07-19, 5:15:09 PM

Done, removing the flagged anecdotal parts
* * *
Author: jw Subject: Sticky Note   Date: 2020-07-19, 5:45:34 PM

We beg to differ slightly on this change. In the text, we try to make a clear distinction between the Installation - an ephemeral work that lasted only ~24 hours - and subsequent exhibitions - derived works allowing interaction with a wider public. The two words may seem to say the same thing from a science perspective, but we do need to respect the terminologies of both the art and the science communities here. Hence we feel that the word exhibition here produces a misalignment with the immediately following text, which is about the installation. Hence we have written "Implementation of the installation" here. However, a more succinct wording could be just "Boundary|Time|Surface: Installation" but we are unsure whether this would satisfy the editor's wishes. Either will work.
* * *
Author: syd     Subject: Sticky Note   Date: 2020-07-18, 3:24:09 PM

DONE
* * *
Author: syd     Subject: Sticky Note   Date: 2020-07-19, 4:09:18 PM

We have rewritten the discussion following the suggestions of the editor. The first part of the discussion starts from the installation and related exhibitions, reviews the audience response, and summarizes these, using the material that was formerly in section 4.5, flagged by the editor for moving to the discussion. The second part of the rewritten discussion relates these results broader questions of interdisciplinary inquiry and how it is perceived, incorporating material from the previous conclusions section that contained references to the literature, as suggested by the editor.
* * *
Author: syd     Subject: Sticky Note   Date: 2020-07-18, 3:24:07 PM

DONE
* * *
Author: syd     Subject: Sticky Note   Date: 2020-07-19, 5:16:04 PM

DONE

Author: Angela Sarao Date: 2020-07-16, 3:54:43 AM

Author: jw Subject: Sticky Note  Date: 2020-07-18, 4:13:28 PM
Abstract reduced to 245 words

Author: Angela Sarao Date: 2020-07-16, 5:55:42 AM

Author: jw Subject: Sticky Note  Date: 2020-07-18, 4:13:51 PM
Moved to introduction

Author: Angela Sarao Date: Indeterminate
Move to Introduction

Author: jw Subject: Sticky Note  Date: 2020-07-18, 3:41:18 PM
Done

[Figure]

Author: Angela Sarao Date: 2020-07-16, 3:55:25 AM

Author: jw Subject: Sticky Note Date: 2020-07-18, 4:15:28 PM
Heading changed to Geological Background

Author: Angela Sarao Date: 2020-07-16, 2:37:46 AM

Author: jw Subject: Sticky Note Date: 2020-07-18, 4:35:49 PM
Strikethrough heading removed, thereby merging sections 2.1 and 2.2 with the introduction as required

Author: Angela Sarao Date: 2020-07-16, 2:37:03 AM

Author: jw Subject: Sticky Note Date: 2020-07-18, 4:36:11 PM
Headings removed, merging 2.1 and 2.2 with the introduction

Author: Angela Sarao Date: 2020-07-16, 2:38:01 AM

Author: jw Subject: Sticky Note Date: 2020-07-18, 4:35:13 PM
Heading removed so as to move this into the introduction

Author: Angela Sarao Date: 2020-07-15, 7:12:03 AM

Author: jw Subject: Sticky Note Date: 2020-07-18, 4:16:36 PM
pp. removed

Author: Angela Sarao Date: 2020-07-16, 2:13:16 AM

Author: jw Subject: Sticky Note Date: 2020-07-18, 4:16:45 PM
pp. removed

Author: Angela Sarao Date: 2020-07-15, 7:13:01 AM

Author: jw Subject: Sticky Note Date: 2020-07-18, 4:19:57 PM
pp. removed

Author: Angela Sarao Date: 2020-07-15, 7:14:52 AM

Author: jw Subject: Sticky Note Date: 2020-07-18, 4:20:35 PM
Highlighted text removed

Author: Angela Sarao Date: 2020-07-16, 2:12:32 AM

Author: jw Subject: Sticky Note Date: 2020-07-18, 4:21:50 PM
(No explanation here: we assume the highlighted text is to be removed)

Author: Angela Sarao Date: 2020-07-16, 2:19:40 AM

Author: jw Subject: Sticky Note Date: 2020-07-18, 4:22:14 PM
Text removed

Author: Angela Sarao Date: 2020-07-16, 9:24:08 AM

Author: jw Subject: Sticky Note Date: 2020-07-18, 4:24:36 PM
No explanation for this flag; we assume the adjacent highlighted text is to be removed

**Page: 5**

Author: Angela Sarao Date: 2020-07-16, 2:20:27 AM

Author: jw  Subject: Sticky Note  Date: 2020-07-18, 4:25:56 PM
Highlighted text removed

Author: Angela Sarao Date: 2020-07-16, 2:20:51 AM

Author: jw  Subject: Sticky Note  Date: 2020-07-18, 4:26:21 PM
We assume this flag relates to the adjacent highlighted text, which we removed

Author: Angela Sarao Date: 2020-07-16, 2:36:49 AM

Author: jw  Subject: Sticky Note  Date: 2020-07-18, 4:37:09 PM
We assume this flag relates to the adjacent heading, which we removed

Author: Angela Sarao Date: 2020-07-16, 2:38:13 AM

Author: jw  Subject: Sticky Note  Date: 2020-07-18, 4:37:21 PM
Heading removed

Author: Angela Sarao Date: 2020-07-16, 3:55:44 AM

Author: jw  Subject: Sticky Note  Date: 2020-07-18, 4:26:36 PM
pp. removed

[Figure]

Author: Angela SaraoDate: 2020-07-15, 7:20:39 AM

Author: jw Subject: Sticky Note  Date: 2020-07-18, 4:26:51 PM
pp. removed

Author: Angela SaraoDate: 2020-07-16, 2:24:19 AM

Author: syd          Subject: Sticky Note  Date: 2020-07-19, 5:47:28 PM
not quite sure what this flag represents. We have removed the pp. in the citation on the previous line

Author: Angela SaraoDate: 2020-07-16, 2:39:26 AM
2. Geological background

Author: jw Subject: Sticky Note  Date: 2020-07-18, 4:40:18 PM
Added (moved from higher up)

Author: Angela SaraoDate: 2020-07-16, 2:53:43 AM

Author: jw Subject: Sticky Note  Date: 2020-07-18, 4:39:49 PM
We assume here that the entire heading is to be deleted, rather than just the number

Author: Angela SaraoDate: 2020-07-16, 3:56:38 AM

Author: syd          Subject: Sticky Note  Date: 2020-07-19, 5:47:45 PM
deleted

Author: Angela SaraoDate: 2020-07-16, 2:40:05 AM

Author: syd          Subject: Sticky Note  Date: 2020-07-19, 5:47:52 PM
deleted

Author: Angela SaraoDate: 2020-07-16, 9:24:34 AM

Author: syd          Subject: Sticky Note  Date: 2020-07-19, 5:48:12 PM
adjacent strikethrough text deleted

Author: Angela SaraoDate: 2020-07-16, 2:42:57 AM

Author: jw Subject: Sticky Note  Date: 2020-07-18, 4:42:33 PM
strikethrough text deleted

Author: Angela SaraoDate: 2020-07-16, 3:08:40 AM

Author: jw Subject: Sticky Note  Date: 2020-07-18, 4:42:44 PM
strikethrough text deleted

Author: Angela SaraoDate: 2020-07-16, 3:08:49 AM

Author: jw Subject: Sticky Note  Date: 2020-07-18, 4:42:51 PM
strikethrough text deleted

Author: Angela SaraoDate: 2020-07-16, 3:08:56 AM

Author: jw Subject: Sticky Note  Date: 2020-07-18, 4:43:16 PM
strikethrough text deleted and wording changed to flow correctly around the deletion

Author: Angela SaraoDate: 2020-07-16, 2:43:31 AM

Author: jw Subject: Sticky Note  Date: 2020-07-18, 4:43:37 PM
strikethrough text deleted

Author: Angela SaraoDate: 2020-07-16, 2:43:52 AM

Comments from page 6 continued on next page

Author: jw Subject: Sticky Note Date: 2020-07-18, 4:43:45 PM
pp. deleted

Author: Angela SaraoDate: 2020-07-16, 2:44:26 AM

Author: jw Subject: Sticky Note Date: 2020-07-18, 4:44:25 PM
strikethrough text deleted

Author: Angela SaraoDate: 2020-07-16, 2:44:49 AM

Author: jw Subject: Sticky Note Date: 2020-07-18, 4:45:20 PM
strikethrough text deleted

Author: Angela SaraoDate: 2020-07-16, 2:45:59 AM

Author: jw Subject: Sticky Note Date: 2020-07-18, 4:45:26 PM
strikethrough text deleted

Author: Angela SaraoDate: 2020-07-16, 2:47:26 AM

Author: jw Subject: Sticky Note Date: 2020-07-18, 4:46:49 PM
strikethrough text deleted

Author: Angela SaraoDate: 2020-07-16, 2:49:22 AM

Author: jw Subject: Sticky Note  Date: 2020-07-18, 4:47:12 PM
strikethrough text deleted

Author: Angela SaraoDate: 2020-07-18, 5:21:22 PM

Author: jw Subject: Sticky Note  Date: 2020-07-18, 4:50:10 PM
These flags are not fully explained but we assume that the headings 2.3.2 and 2.3.3 are to be removed following the instruction to remove 2.3.1.

Author: Angela SaraoDate: 2020-07-16, 2:53:18 AM

Author: jw Subject: Sticky Note  Date: 2020-07-18, 5:22:12 PM
Strikethrough text deleted: we assume following the deletion of 2.3.1 that the entire heading must be deleted and not just the number

Author: Angela SaraoDate: 2020-07-16, 9:24:41 AM

Author: jw Subject: Sticky Note  Date: 2020-07-18, 5:22:52 PM
Strikethrough text deleted; heading also deleted as explained above

Author: Angela SaraoDate: 2020-07-16, 2:55:44 AM

Author: Angela SaraoDate: 2020-07-16, 2:59:22 AM

: implementation of the exhibition

Author: jw  Subject: Sticky Note  Date: 2020-07-19, 5:49:11 PM

We beg to differ slightly on this change.  In the text, we try to make a clear distinction between the Installation - an ephemeral work that lasted only ~24 hours - and subsequent exhibitions - derived works allowing interaction with a wider public. Hence we feel that the word exhibition here produces a misalignment with the immediately following text, which is about the installation.  Hence we have written "Implementation of the installation" here.  However, a more succinct wording could be just "Boundary|Time| Surface: Installation" but we are unsure whether this would satisfy the editor's wishes. We would be ok with either.

Author: Angela SaraoDate: 2020-07-16, 2:58:38 AM

Author: jw  Subject: Sticky Note  Date: 2020-07-18, 5:21:10 PM

Strikethrough text deleted

Author: Angela SaraoDate: 2020-07-16, 3:02:27 AM

Author: jw  Subject: Sticky Note  Date: 2020-07-18, 5:03:47 PM

strikethrough text deleted; however, this required us to change 'use' to 'used'

Author: Angela SaraoDate: 2020-07-16, 3:01:51 AM

Author: jw  Subject: Sticky Note  Date: 2020-07-18, 5:01:46 PM

We deleted the adjacent text

Author: Angela SaraoDate: 2020-07-16, 3:02:44 AM

Author: jw  Subject: Sticky Note  Date: 2020-07-18, 5:01:19 PM

strikethrough text deleted

Author: Angela SaraoDate: 2020-07-19, 5:49:31 PM

Author: jw  Subject: Sticky Note  Date: 2020-07-18, 5:04:22 PM

We are unsure what is being flagged here, but have made the text more succinct

Author: Angela SaraoDate: 2020-07-16, 3:13:54 AM

Author: syd          Subject: Sticky Note  Date: 2020-07-19, 5:51:44 PM

strikethrough text deleted

**Page: 9**

[Figure]

Author: Angela Sarao Date: 2020-07-16, 3:05:51 AM

Author: jw Subject: Sticky Note Date: 2020-07-18, 5:12:38 PM
We are unsure what was being flagged here, as there is no direct explanation. We have edited the flagged paragraph to make it more concise and factual.

Author: Angela Sarao Date: 2020-07-16, 3:16:12 AM

Author: jw Subject: Sticky Note Date: 2020-07-18, 5:28:35 PM
Strikethrough text deleted

Author: Angela Sarao Date: 2020-07-16, 3:06:55 AM

Author: jw Subject: Sticky Note Date: 2020-07-18, 5:29:12 PM
Title changed as required

Author: Angela Sarao Date: 2020-07-16, 3:06:24 AM

Author: jw Subject: Sticky Note Date: 2020-07-18, 5:28:58 PM
Title changed as required

Author: Angela Sarao Date: 2020-07-16, 3:24:11 AM

Author: syd Subject: Sticky Note Date: 2020-07-19, 5:53:53 PM
We assume this flag relates to the general restructuring under point 10

Author: Angela SaraoDate: 2020-07-16, 3:22:29 AM

Author: jw Subject: Sticky Note  Date: 2020-07-18, 5:31:56 PM
Strikethrough text deleted

Author: Angela SaraoDate: 2020-07-16, 3:22:36 AM

Author: jw Subject: Sticky Note  Date: 2020-07-18, 5:31:31 PM
Strikethrough text deleted, and adjacent text shortened to restore the flow.

Author: Angela Sarao Date: 2020-07-16, 3:30:14 AM

Author: jw Subject: Sticky Note Date: 2020-07-18, 5:41:15 PM
These two flags contain no explanation but we presume they relate to the questions mentioned in the abstract. We have moved mention of the questions from the abstract to here, so as to keep within the recommended abstract length and at the same time comply with comment 10.

Author: Angela Sarao Date: 2020-07-16, 3:51:46 AM

Author: jw Subject: Sticky Note Date: 2020-07-18, 5:41:11 PM
These two flags contain no explanation but we presume they relate to the questions mentioned in the abstract. We have moved mention of the questions from the abstract to here, so as to keep within the recommended abstract length and at the same time comply with comment 10.

Author: Angela Sarao Date: 2020-07-18, 5:43:50 PM

Author: jw Subject: Sticky Note Date: 2020-07-18, 5:45:39 PM
These two flags contain no text, but we presume they relate to point 11 on discussion, where it says "there are some sparse considerations about results in the other sections that should be moved here". Accordingly, we have moved this to the discussion.

Author: Angela Sarao Date: 2020-07-16, 3:33:49 AM

Author: syd Subject: Sticky Note Date: 2020-07-19, 5:54:53 PM
Title changed and section moved as recommended

Author: Angela Sarao Date: 2020-07-18, 5:42:24 PM

Author: syd Subject: Sticky Note Date: 2020-07-19, 5:56:21 PM
We have moved the whole subsection into the discussion as recommended.

**Page: 12**

Author: jw Subject: Sticky Note  Date: 2020-07-19, 6:02:55 PM
I can't make Acrobat correctly indicate the location of this highlight; I'm hoping it's something I've dealt with.

Author: Angela SaraoDate: 2020-07-16, 7:22:01 AM

Author: jw Subject: Sticky Note  Date: 2020-07-18, 6:25:24 PM

Author: Angela SaraoDate: 2020-07-18, 6:18:49 PM
Revise

Author: jw Subject: Sticky Note  Date: 2020-07-18, 6:22:25 PM
In the revised discussion we begin with the points formerly made in section 4.5. We have removed the section on connecting colonial world-views with the history of geology, as the connection of this section to the exhibition was predominantly artistic rather than scientific. We have also moved the section on global distribution of stratotypes to the introduction as suggested.  Finally, we have incorporated in the revised discussion the points from the conclusions that included cited references.

Author: Angela SaraoDate: 2020-07-16, 3:39:06 AM

Author: jw Subject: Sticky Note  Date: 2020-07-18, 6:15:57 PM
Removed

Author: Angela SaraoDate: 2020-07-18, 6:23:51 PM
??

Start from the exhibition and then extract principles, relationships, or generalisations.

How all these correlate with your exibition?

Author: jw Subject: Sticky Note  Date: 2020-07-19, 6:02:50 PM
Flagged text deleted, along with some following closely related sentences.   They reflect perspectives that we have become more aware of as authors through our joint work. We will find another venue to make these points.

[Figure]

Author: Angela Sarao Date: 2020-07-16, 7:22:36 AM

Author: jw Subject: Sticky Note Date: 2020-07-18, 6:24:30 PM
Moved to earlier section following editor direction

Author: Angela Sarao Date: 2020-07-16, 3:58:23 AM

Author: jw Subject: Sticky Note Date: 2020-07-18, 6:26:49 PM
removed

Author: Angela Sarao Date: 2020-07-18, 5:49:14 PM
A great part of this paragraph can be moved in the Background subsection about stratotypes.
Here you should discuss YOUR results, or in case the link beween your exibition and the choice of a GSSP

Author: jw Subject: Sticky Note Date: 2020-07-18, 6:24:55 PM
moved as suggested

Author: Angela Sarao Date: 2020-07-18, 6:29:19 PM

Author: jw Subject: Sticky Note Date: 2020-07-18, 6:29:34 PM
pp.deleted

Author: Angela Sarao Date: 2020-07-16, 3:48:12 AM

Author: jw Subject: Sticky Note Date: 2020-07-18, 6:27:26 PM
pp. deleted

Author: Angela Sarao Date: 2020-07-18, 6:27:26 PM

Author: syd Subject: Sticky Note Date: 2020-07-19, 6:00:31 PM
Material moved from Conclusions to Discussion where it contains references as recommended

**Page: 14**

[Figure]

Author: Angela Sarao Date: 2020-07-18, 6:27:30 PM

Author: jw Subject: Sticky Note Date: 2020-07-18, 6:29:56 PM
These editor flags contain no specific comments. We have made the conclusions more succinct by removing material with citations to the discussion and incorporating some general points from the discussion here

Author: Angela Sarao Date: 2020-07-16, 8:20:22 AM
some sentences of the conclusion should be moved to the Discussion, mainly those that refers to others' works.

Author: jw Subject: Sticky Note Date: 2020-07-18, 6:27:59 PM
Sentences referring to others works moved to discussion

Author: Angela Sarao Date: 2020-07-16, 3:49:21 AM

Author: syd Subject: Sticky Note Date: 2020-07-19, 5:56:51 PM
pp removed from citation

Author: Angela Sarao Date: 2020-07-16, 3:49:35 AM

Author: syd Subject: Sticky Note Date: 2020-07-19, 5:57:03 PM
pp removed from citation

Author: Angela Sarao Date: 2020-07-16, 7:59:32 AM

Author: syd Subject: Sticky Note Date: 2020-07-19, 5:59:28 PM
Our version of Acrobat is not revealing the location of this flag, though it is listed as a comment; we hope it is something we have covered.

Revised manuscript with tracked changes displayed

**Boundary|Time|Surface: Assessing a meeting of art and geology through an ephemeral sculptural work**

Sydney A. Lancaster[1], John W.F. Waldron[2]

[1]Edmonton, Alberta, T6E1G6, Canada
[2]Earth and Atmospheric Sciences, University of Alberta, Edmonton, T6G2E3, Canada

Sydney A. Lancaster: ORCID: 0000-0002-5843-3947
John W.F. Waldron: ORCID: 0000-0002-1401-8848

*Correspondence to*: Sydney A. Lancaster (sydneylancaster@ualberta.net)

20
25

**Abstract.** *Boundary|Time|Surface* was an ephemeral site-specific sculpture created to draw attention to the construction of social, political, scientific and aesthetic boundaries that divide the Earth. One such practice is the scientific subdivision of geologic time. The sculpture comprised a 150 m fence along the international stratotype separating Ordovician from Cambrian strata in Gros Morne National Park, Canada. The fence was constructed by hand in one day, on a falling tide, from materials found on site, with minimal environmental impact. During the following tidal cycles, it was dismantled by wave and tide action. This cycle of construction and destruction was documented with time-lapse photography and video, and brought to the public through exhibitions, public talks, and a book. Exhibitions derived from the documentation of ephemeral works function as translations of the original experience. In this case, they provided opportunities for public interaction with media that served both as aesthetic objects, and as sources of information about the site's geological and socio-political history. We assess the role of the installation and its documentation in drawing public attention to boundaries, and examine responses including attendance records and written visitor comments as indications of viewers' engagement with the concepts presented. Of several thousand visitors to exhibitions, 418 written comments reflected the viewers' engagement with both the location and the underlying concepts. Both the original installation and the subsequent work allowed audiences to explore human understanding and acquisition of knowledge about the Earth, and how world-views inform the process of scientific inquiry.

**Moved (insertion) [2]**

**Moved down [1]:** Geologists and artists have taken different

**Moved up [2]:** During the remainder of the tidal cycle, and those

**Moved down [3]:** *Boundary|Time|Surface* attempted to convey

**Moved (insertion) [3]**

[revised manuscript text omitted]

---

## Author Response (AR3)

Editors, Geoscience Communication

2020-07-23

Dear Drs Saraò and Illingworth

Thank you for your comments and changes required for the finalization of our paper: "Boundary|Time|Surface: Assessing a meeting of art and geology through an ephemeral sculptural work". We have completed the requested changes, renumbering figures where necessary and changing the layout of the figures. We attach below a version of text document with changes tracked. The results of these changes will be shown in the revised 'clean' manuscript and figures pdf. There is also a new version of the supplement, to make sure all the cross-references are correct and as required.

Yours,

John Waldron (for Sydney Lancaster)

[revised manuscript text omitted]